# Synthesis of D-π-A′-π-A Chromophores with Quinoxaline Core as Auxiliary Acceptor and Effect of Various Silicon-Substituted Donor Moieties on Thermal and Nonlinear Optical Properties at Molecular and Material Level

**DOI:** 10.3390/molecules28020531

**Published:** 2023-01-05

**Authors:** Alexey A. Kalinin, Liliya N. Islamova, Sirina M. Sharipova, Guzel M. Fazleeva, Alexey A. Shustikov, Adel I. Gaysin, Artemiy G. Shmelev, Anastasiya V. Sharipova, Tatyana A. Vakhonina, Olga D. Fominykh, Olga B. Babaeva, Ayrat R. Khamatgalimov, Marina Yu. Balakina

**Affiliations:** 1Arbuzov Institute of Organic and Physical Chemistry, FRC Kazan Scientific Center, Russian Academy of Sciences, Arbuzov Str. 8, 420088 Kazan, Russia; 2Zavoisky Physical-Technical Institute, FRC Kazan Scientific Center, Russian Academy of Sciences, Sibirsky Tract 10/7, 420029 Kazan, Russia

**Keywords:** D-π-A′-π-A chromophores, quinoxaline, silicon-substituted donor, NLO activity, second harmonic generation

## Abstract

Novel D-π-A′-π-A chromophores with quinoxaline cores as auxiliary acceptors and various donor moieties (aniline, carbazole, phenothiazine, tetrahydroquinoline) containing bulky *tert*-butyldimethylsilyloxy (TBDMSO) groups and tricyanofuranyl (TCF) acceptors with bulky cyclohexylphenyl substituents were synthesized via eight- to nine-step procedures, and their photo-physical and thermal properties were investigated. The values of the chromophores’ first hyperpolarizabilities were calculated in the framework of DFT at the M06-2X/aug-cc-pVDZ computational level; the effect of the introduction of the TBDMSO group into the donor fragment is shown to be inessential, as this group is not coupled to the π-conjugated system of the chromophore. The chromophore with the tetrahydroquinoline donor has a first hyperpolarizability value of 937 × 10^−30^ esu, which is the highest for the studied chromophores. Atomistic modeling of composite materials with the studied chromophores as guests demonstrated that the presence of bulky substituent in the donor fragment prevents notable aggregation of chromophores, even at high chromophore content (40 wt.%). The nonlinear optical performance of guest–host materials with 25 and 40 wt.% of suggested chromophore content was studied using a second harmonic generation technique to give the NLO coefficient, *d*_33_ up to 52 pm/V.

## 1. Introduction

Two main applications—medicinal and technological—stimulate the interest of researchers in the synthesis and investigation of the properties of complex molecules with quinoxaline core. Quinoxalines and fused quinoxalines possess a wide range of biological activities such as antitumoral, antibacterial and antiviral activity [1,2,3,4]. These compounds have found technical applications as luminescent materials [5,6], components for optoelectronics [6] and organic photovoltaics [7,8,9]. The combination of the quinoxaline core with aromatic/heteroaromatic or ethylene/acetylene moieties in one molecule provided valuable photo-physical and electro-chemical properties. For example, dipyrrolyl-, diquinolinyl-quinoxalines [10,11,12,13]; diquinoxalinylbiindolizines [14,15,16,17] and dicarbazolyl-, di(biphenyl)-, tetrakis(pyridinyl)phenyl-quinoxalines [18,19,20] exhibit sensor ability for anions and cations, redox-switched binding of metal cations and yellow/white/blue luminescence, correspondingly. Various aminostyrylquinoxaline derivatives demonstrate luminescent properties along with halochromism [21,22] or mechanofluorochromism [23] and gelation ability [24]. Depending on the site of attachment of the aminostyryl moiety to the quinoxaline core, it is possible to adjust the luminescent characteristics [25] and dye-synthesized solar cells’ performances [26].

One of the new directions in the field of quinoxaline dyes is the study of their nonlinear optical (NLO) properties, both at the molecular [27,28] and at the material level [29,30]. The extensive investigations of the structure–property relationship for push–pull NLO chromophores used in the development of organic polymer materials for photonic applications resulted in the formulation of the main guidelines for the design of such molecular systems. One of the key features of the design of efficient NLO chromophores is the introduction of bulky groups into various structural fragments of the chromophore to prevent detrimental electrostatic interactions resulting in the deterioration of chromophores’ acentric ordering, which is necessary for the quadratic NLO response [31]. *tert*-Butyldimethylsilyloxy (TBDMSO) or *tert*-butyldiphenyllsilyloxy (TBDPSO) groups are often used for this purpose [32,33,34,35]. These bulky groups are introduced to increase chromophore solubility and improve film-forming ability; as these groups reduce electrostatic interaction between chromophores, chromophore number density of the material may be essentially increased up to receiving monolithic (neat) chromophore films [32,35]. The main gain from the introduction of such groups in chromophore design is the improvement of poling efficiency due to the realization of larger accessible space for chromophore reorientation, enhancing its mobility [32,36], which results in high EO coefficient and thermal stability [32,37,38].

Here, we study the effect of bulky TBDMSO groups introduced into various chromophore aniline donor fragments through ethylene or hexylene spacers on thermal and linear properties of chromophores and (non)linear optical properties of PMMA-based composite polymer materials with the proposed chromophores as guests.

## 2. Results and Discussion

### 2.1. Synthesis

Starting from commercially available *N*-ethylaniline, *N*-ethyl-*N*-(2-hydroxyethyl)aniline, *N,N*-di(2-hydroxyethyl)anylnine, carbazole and phenothiazine, the desired chromophores were synthesized via eight- to nine-step procedures, as shown in Figure 1. At the beginning, key compounds **2a–d**, **2e′**, **2f′**, **3**, **3′** and **8** were obtained. The first of them—monosubstituted olefins **2a–d** with aminobenzene or heterocyclic moieties—were synthesized in three or four steps, including hydroxyalkylation of the corresponding amines, subsequent acylation with acetic anhydride, Vilsmeier–Haack formylation and the final Wittig reaction. In the case of the Wittig reaction of phenothiazine derivatives **1e,** hydroxy derivative **2e′** was obtained instead of the acetoxy derivative. Palladium-catalyzed Heck reactions of olefins **2a–d,e′** and the second key compound—6-bromo-2-methyl-3-phenylquinoxaline (**3**), obtained in turn from *o*-phenylenediamine and 1-phenylpropane-1,2-dione and followed by the release of the necessary amphi-methylbromoisomeric derivative **3** [39]—led to *trans*-1,2-disubstituted olefins **4a–d** and **5e**. The replacement of the acetoxy group with TBDMSO was carried out in two steps through hydrolysis of acetyl derivatives **4a–d** with the formation of alcohols **5a–d**. The alkylation of compounds **5a–e** using TBDMS-Cl resulted in compounds **6a–e**. Riley oxidation of the methyl group at position 2 of the quinoxaline core of olefins **6a–e** using selenium dioxide led to aldehydes **7a–e**. For the transformation of alkyl groups into acyl groups in quinoxalinone derivatives, there are other oxidative procedures, for example, with the use of chromium anhydride (VI) [40,41].

**Scheme 1 molecules-28-00531-sch001:**
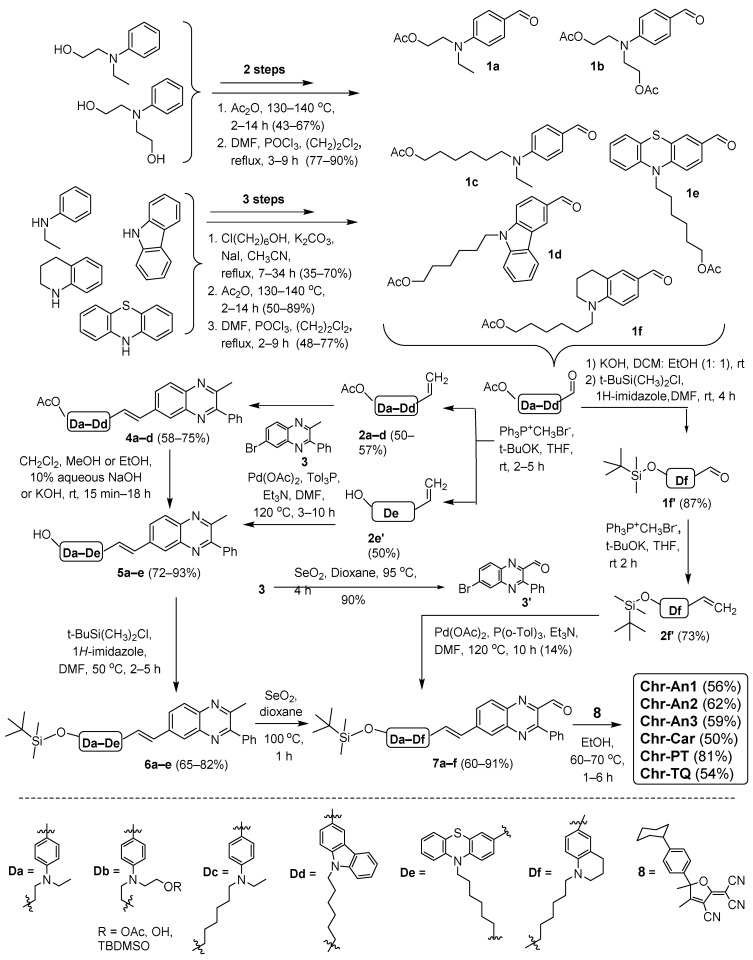
Synthetic approaches to quinoxaline-based NLO chromophores with various donors; the structure of **Chr-An1**, **Chr-An2**, **Chr-An3**, **Chr-Car**, **Chr-PT** and **Chr-TQ** is shown in Figure 1.

The synthetic approach given here providing aminostyrylquinoxalinylcarbaldehydes **7a–e** with the Riley reaction at the last step made it possible to obtain a wide range of their derivatives, both with a dialkylaniline donor moiety [29,42] and with a carbazole or phenothiazine moiety. However, it transpired to be difficult to obtain aldehyde **7f** using this approach due to the low conversion (~30%) at the final step. Furthermore, close *R_f_* values of the product and the starting reagent made it difficult to isolate **7f** using column chromatography. To obtain the aldehyde **7f**, a slightly different synthetic approximation was implemented: olefin **2f′** and quinoxalinecarbaldehyde **3′**, synthesized from tetrahydroquinoline via a six-step procedure and the oxidation of quinoxaline **3**, respectively, were used in the Heck reaction. Final Knoevenagel condensation of the aldehyde group in compounds **7a–f** and the methyl group in the third key compound **8** [42] led to target chromophores under mild base-free conditions. All disubstituted olefin derivatives **4**, **5**, **6** and **7** were isolated as E-isomers, as shown in Figure 1, and chromophores were isolated as E,E-isomers, as evidenced by ^1^H NMR (*J*_-CH=CH-_ = ~16 Hz).

The signals from ortho- and meta-protons of phenyl group at quinoxaline moieties were shifted to the higher field and para-proton of phenyl group resonate in the lower field (7.60–7.53 ppm) due to the shielding effect of the aryl substituent in TCF moieties. This indicates that, in the chloroform solution, only one conformer for all compounds existed with close spatial arrangement of Ph and CyPh moieties, as shown in Appendix A.

### 2.2. Linear Optical Properties

An intense intramolecular charge transfer (ICT) band in the visible region in the range of 536–665 nm, depending on the solvent polarity, is characteristic of all chromophores studied (Figure 2). All chromophores are characterized by a bathochromic shift in the absorption maximum when going from nonpolar solvents, such as dioxane, to those of moderate polarity, such as chloroform or dichloromethane—positive solvatochromism (chloroform/dioxane). Further increase in solvent polarity (acetonitrile) results in negative solvatochromism (acetonitrile/chloroform), the value of hypsochromic shift being greater than that of bathochromic shift. As can be seen from Table 1, this is a general trend for chromophores with a quinoxaline core in the π-bridge regardless of the type of donor moiety, which distinguishes this class of chromophores from chromophores with a thiophene core in the π-bridge [38]. In accordance with the value of hypsochromic shift, the studied chromophores may be arranged in the following series in all solvents: λ_max_(**Chr-TQ) >** λ_max_(**Chr-An3**) > λ_max_(**Chr-An1**) > λ_max_(**Chr-An2**) > λ_max_(**Chr-PT**) > λ_max_(**Chr-Car**). In the case of **Chr-An1** and **Chr-An2** with substituent with the ethylene spacer, the closeness of an electro-negative oxygen atom (or two atoms) to aniline nitrogen leads to some weakening of the donor moiety and a hypsochromic shift in the absorption maximum in comparison with **Chr-An3** with the hexylene spacer in the substituent. The incorporation of an additional benzene moiety into the donor also leads to a hypsochromic shift in the absorption maximum of carbazole-based chromophore **Chr-Car** in comparison with chromophore **Chr-An3** up to 74 nm (0.25 eV). 

However, the incorporation of a sulfur atom does not lead to a noticeable shift in the absorption maximum for phenotiazine-based chromophore **Chr-PT** in comparison with chromophore **Chr-Car**, but this leads to a weakening of the ICT band, a decrease in the molar extinction coefficient by 20% and the complication of the spectrum—the appearance of new absorption bands in the short-wave region. It is interesting to note that the chromophore **Chr-PT** exhibits a significant negative solvatochromic shift (0.20 eV). D-π-A chromophore salts are known to demonstrate a negative solvatochromic effect exceeding this value [44,45]. In contrast, a decrease in the pyramidality of the aniline nitrogen, due to the binding of the nitrogen atom with the benzene part of the aniline donor through the propylene spacer—in the composition of the tetrahydropyridine ring rather than the addition of a benzene ring (as in the **Chr-Car** chromophore)—leads to a bathochromic shift in the absorption maximum of the **Chr-TQ** chromophore in comparison with **Chr-Car**, which reaches almost 100 nm in solvents with moderate polarity (Table 1). A bathochromic shift (up to 20 nm) in the absorption maximum is observed when comparing **Chr-TQ** with a similar chromophore **DBA-VQV-TCF_CyPh_** with the same acceptor and π-bridge, but with a dibutylaniline donor [30].

### 2.3. Thermal Properties

The thermal stability of chromophores was investigated using simultaneous TG/DSC analysis. Figure 3 and Figure 4 show the TG and DSC curves of the studied chromophores. The studied quinoxaline-based chromophores exhibit similar characteristics of weight loss and have high thermal stability; the decomposition temperatures, T_d_, at which 5% mass loss occurs at heating are above 259 °C (Table 2). 

Previously [29], we found that thermal stability for quinoxaline-based chromophores estimated using TGA is somewhat overestimated due to the occurrence of decomposition without weight loss; thus, DSC seems to provide more reliable determination of thermal stability for this class of compounds. For **Chr-An3** with a TBDMSO group and the hexylene spacer, TGA and DSC techniques provide notably different estimations of thermal stability, similar to the case of earlier-studied quinoxaline-based chromophores with dibutylaniline donors, while for **Chr-An1** and **Chr-An2** with the substituent containing the ethyl spacer, the difference in the obtained values of thermal stability estimated by the two methods is much smaller (Figure 4). Changing the aniline donor for a heterocyclic donor (chromophores **Chr-Car** and **Chr-PT**), on the other hand, results in close values of thermal stability obtained using TGA and DSC. T_d_, obtained using DSC, is higher for these two chromophores compared to **Chr-An1** and **Chr-An3,** in spite of their similar melting temperatures (Table 2). The difference between the melting temperature and T_d_ is about 50 °C for **Chr-Car** and **Chr-PT**, while for **Chr-An1** and **Chr-An3** they are 20 and 34 °C, respectively. In the case of **Chr-An2,** high thermal stability seems to be conditioned by its high mp—just after melting, the chromophore decomposes. The mentioned chromophores appeared to be crystalline compounds with mp above 192 °C (Table 2 and Figure 4a–e). The DSC curve for **Chr-TQ** is more complex; there are two endo- and two exothermic peaks. As the first exothermic peak is low-intensive (close to the base line), it is not clear whether the corresponding temperature is the T_d_. To clarify this, the chromophore **Chr-TQ** was heated up to 180 °C inside the DSC/TGA unit and the complete chromophore decomposition was confirmed using TLC. Thus, **Chr-TQ** manifests much lower thermal stability compared to that of all five other chromophores studied here, with the difference in T_d_ reaching almost 100 °C compared to the most stable chromophores (**Chr-Car**, **Chr-PT**, **Chr-An2**).

### 2.4. Quantum-Chemical Calculations and Molecular Modeling

A conformational search was used to determine the most stable conformers of the studied chromophores; in all cases except **Chr-PT,** these were *tct* conformers (Appendix A). The values of electric characteristics for this conformer are presented in Table 3. For the reference **DBA-VQV-TCF_CyPh_** (Appendix A) without TBDMSO-containing substituents, the *cct* conformer was found to be the most stable [30]; however, the difference in energy between *tct* and *cct* conformers does not exceed 1 kcal/mole and the values of first hyperpolarizability do not differ notably (within ~4%), which is in agreement with the regularity obtained earlier for the chromophores with quinoxaline core [27]. The difference in first hyperpolarizability values for various conformers is somewhat higher for **Chr-PT**—it reaches 10%. The dihedral angles characterizing the geometry of the *tct*-conformers of the chromophores (Appendix A) are given in Table 4. 

Comparison of the β_tot_ values given in Table 3 with those of **DBA-VQV-TCF_CyPh_** (805·10^−30^ esu) demonstrates that the introduction of bulk TBDMSO groups into the donor fragment does not essentially affect the values of first hyperpolarizability; for **Chr-An1**, **Chr-An2** and **Chr-An3,** the difference is equal to 1%, 13% and 1%, respectively. In fact, the substituent is not involved in the π-conjugated system of the chromophore, i.e., in the NLO active region of the molecular space. In the case of **Chr-Car** and **Chr-PT** with the donor fragments being heterocyclic fused systems, β_tot_ is essentially smaller (less than 1.6 and 1.7 times in comparison with **DBA-VQV-TCF_CyPh_**). The value of β_tot_ for **Chr-TQ** is 1.2 times higher than that of **DBA-VQV-TCF_CyPh_**, i.e., the tetrahydroquinoline donor is close in efficiency to the aniline donors of **Chr-An2** and **Chr-An3**).

Molecular modeling in amorphous cells (Appendix A) for composite chromophore/PMMA materials demonstrated that the introduction of bulky substituents containing TBDMSO fragments into the aniline donors of chromophores with divinylquinoxaline bridges and TCF_CyPh_ acceptors prevents pronounced chromophore aggregation in polymer matrixes; the maximal size of clusters formed via π–π stacking interactions are not greater than four units, even at high (40 wt.%) chromophore content (Table 5). When the hexylene spacer is used instead of the ethylene spacer (**Chr-An3** and **Chr-An1**)**,** the portion of bound chromophores does not exceed 32% at a chromophore content of 40 wt.%, and the formed clusters are dimers (Table 5). The introduction of two substituents with the TBDMSO group (**Chr-An2**) results in better isolating ability than in the case of one substituent (**Chr-An1**): 35% of chromophores are noncovalently bound and the maximal cluster size is three. When the donor fragment is fused heterocycle (**Chr-Car, Chr-PT** and **Chr-TQ**), a notably greater portion of chromophores are involved in noncovalent bonding than in the case of **Chr-An3** with a similar substituent (TBDMSO with the hexylene spacer) and the size of the cluster differing from 3 to 5. Thus, the introduction of two TBDMSO-containing substituents with the short ethylene spacer provides a similar isolating effect to one substituent with a long spacer (the hexylene spacer) at the same aniline donor.

### 2.5. Experimental NLO Activity of Composite Polymer Materials Doped by Chromophores

Thin polymer films of composite materials doped with the synthesized chromophores were fabricated and poled in a corona discharge field. Their characteristics (UV-vis spectra before and after poling, poling temperature, order parameter and film thicknesses) are given in Appendix A. The NLO coefficient of the sample *d*_33,*s*_ was estimated as follows [46]: $\frac{d_{33,s}}{d_{11,q}}=\sqrt{I_{s}/I_{q}}\frac{l_{c,q}}{l_{s}}F,$ where *I_s_* and *I_q_* are SHG intensities produced by the sample and the quartz, respectively, and measured in the same configuration, *l_c,q_* is quartz coherence length related to 1028 nm (calculated as 13 µm), *l_s_* is sample thickness and *F* is correction factor (1.2 when $l_{c,q}\gg l_{s}$) [47]. The values of the NLO coefficient of poled PMMA-based polymer films doped with 25 wt.% of **Chr-An1**, **Chr-An2**, **Chr-An3**, **Chr-Car**, **Chr-PT** and **Chr-TQ** were measured to be in the range 23–46 pm/V (Figure 5). The increase in chromophore load from 25 to 40 wt.% in the cases of chromophores with aniline donors led to the increase in *d*_33_ values up to 52 pm/V for **Chr-An3/PMMA**. In the case of chromophores with heterocyclic donors (**Chr-Car/PMMA**, **Chr-PT/PMMA** and **Chr/TQ/PMMA**), the growth in chromophore load left *d*_33_ unchanged. A high value of *d*_33_ for **Chr-An3**(40)**/PMMA** is in accordance with the conclusion based on molecular modeling concerning good isolating ability of bulky substituents with the hexylene spacer. Three materials, **Chr-An1**(25)**/PMMA**, **Chr-An3**(25)**/PMMA** and **Chr-Car**(25)**/PMMA,** exhibit close values of *d*_33_ 44–46 pm/V. The composite material **Chr-PT**(25)**/PMMA**, doped with the chromophore with heterocyclic sulfur-containing donor, is characterized by a smaller (by ~40%) *d*_33_ value 33 pm/V. The smaller *d*_33_ value for **Chr-An2/PMMA** seems to be due to the worse film-forming ability of the material caused by the worse solubility of the chromophore. Thus, the introduction of the TBDMSO group permits increasing the chromophore content in the material with the growth in NLO response. Relatively close *d*_33_ values at 25 and 40 wt.% of chromophore content give grounds to predict that optimal chromophore load is intermediate between them. Further modification of chromophore structure may lead to higher values of *d*_33_ at high chromophore load. It is interesting to note close values of *d*_33_ for composite materials doped with **Chr-Car** and **Chr-An3** chromophores at similar chromophore content, in spite of an almost two-fold difference in their β_tot_ values (497 × 10^−30^ and 798 × 10^−30^ esu, respectively). A rather high value of *μβ* product for **Chr-Car** in combination with rather high NLO coefficient of composite chromophore-containing material at optical transparency in near-IR regions make these materials promising candidates for use in optical modulators, which are necessary for the development of short-range local networks.

The dependences of the normalized SHG signal on the laser beam incidence angle for some composite polymer films are given in Figure 6.

The polymer films **Chr-An1/PMMA, Chr-An3/PMMA, Chr-Car/PMMA** and **Chr-PT/PMMA** demonstrate rather high long-term stability of the NLO response—their *d*_33_ values are preserved at 90–95% during 6–12 months at room temperature.

## 3. Materials and Methods

### 3.1. General

The IR, NMR spectra and ESI mass spectra were registered using the equipment of the Assigned Spectral-Analytical Center of FRC Kazan Scientific Center of RAS. NMR experiments were performed with Bruker AVANCE-600, AVANCE-500 and AVANCE-400 (600 MHz, 500 MHz and 400 MHz for ^1^H NMR, 150 MHz, 125 MHz and 100 MHz for ^13^C NMR) spectrometers. Chemical shifts (*δ* in ppm) are referenced to the solvents. IR spectra were recorded using a Bruker Vector-22 FT-IR spectrometer. High-resolution ESI mass spectra (HRMS (ESI)) were obtained using an Impact II (Bruker Daltonik GmbH, Bremen, Germany) mass spectrometer with an Elute UHPLC (Bruker Daltonik GmbH, Bremen, Germany) LC system. The column used was a YMC-Triart C18 (50 × 2.0 mm; 3 μm) with a flow rate of 0.3 mL/min. Analytes were ionized using electrospray in positive polarity. ESI conditions were set with the capillary temperature at 220 °C, capillary voltage at −3.5 kV and a sheath gas flow rate of 8 L/min. UV–vis spectra were recorded at room temperature using a UV-6100 ultraviolet/visible spectrophotometer using 10 mm quartz cells. Spectra were registered with a scan speed of 480 nm/min, using a spectral width of 1 nm. All samples were prepared in solutions with a concentration of ~3 × 10^−5^ mol/L. The melting points, mp, for new compounds in the experimental section were determined using a melting point meter MF-MP-4. The thermal stabilities and mp of chromophores were investigated through simultaneous thermal analysis (thermogravimetry/differential scanning calorimetry—TG/DSC) using a NETZSCH (Selb, Germany) STA449 F3 instrument. Approximately 3–4 mg samples were placed in an Al crucible with a pre-hole in the lid and heated from 30 to 500 °C. The same empty crucible was used as the reference sample. High-purity argon was used with a gas flow rate of 50 mL/min. TG/DSC measurements were performed at the heating rates of 10 K/min. The thickness of doped polymer films was determined through the AFM technique using a dimension FastScan high-resolution scanning probe microscope (Bruker, Germany). Ultra-sharp silicon probes Bruker ScanAsyst-air with a tip curvature radius of ~2 nm were used. Organic solvents used were purified and dried according to standard methods. The reaction progress and the purity of the obtained compounds were controlled using TLC on Sorbfil UV-254 plates with visualization under UV light. Compounds **1b** [48], **1c** [49], **1f** [50], **3** [39], **8** [42], **Chr-An1** and **Chr-Car** were synthesized according to the literature [43].

### 3.2. 6-(10H-Phenothiazin-10-yl)hexan-1-ol

A mixture of 10H-phenothiazine (2.00 g, 0.01 mol), 6-chlorohexan-1-ol (2.06 g, 0.015 mol), potassium carbonate (2.77 g, 0.02 mol) and sodium iodide (3.00 g, 0.02 mol) in dry 20 mL CH_3_CN was refluxed for 34 h. The reaction mixture was cooled, poured into water and extracted with CH_2_Cl_2_. The organic layer was separated, washed with water, dried over anhydrous MgSO_4_ and filtered. The solvent was removed at reduced pressure, and the residue was purified using silica gel column chromatography (eluent petroleum ether—EtOAc, gradient from 50:1 to 10:1) to give the title product. Yield (1.05 g, 35%), colorless oil, *R_f_* 0.22 (1:0.3 hexane/EtOAc). IR (KBr, *ν*_max_/cm^−1^): 3346 (OH), 2932 (CH), 2857 (CH), 1594 (C-N, C=C), 1571, 1457, 1334, 1250, 1229, 1182, 1127, 1106, 1039, 928, 855. ^1^H NMR (400 MHz, CDCl_3_): *δ* 7.17–7.13 (m, 4H), 6.93–6.89 (m, 2H), 6.87–6.85 (m, 2H), 3.85 (t, *J* = 7.1 Hz, 2H, NCH_2_), 3.59 (t, *J* = 6.5 Hz, 2H, CH_2_OH), 1.85–1.78 (m, 2H, NCH_2_(CH_2_)_3_CH_2_CH_2_OH), 1.58–1.35 (m, 7H, NCH_2_(CH_2_)_3_CH_2_CH_2_OH). ^13^C NMR (100 MHz, CDCl_3_) *δ* 145.3 (C), 127.4 (CH), 127.1 (CH), 125.0 (C), 122.3 (CH), 115.4 (CH), 62.7 (CH), 47.2 (CH), 32.6 (CH), 26.8 (CH), 26.6 (CH), 25.3 (CH). 

### 3.3. 6-(10H-Phenothiazin-10-yl)hexyl Acetate

A mixture of 6-(10*H*-phenothiazin-10-yl)hexan-1-ol (350 mg, 1.17 mmol) and acetic anhydride (119 mg, 1.17 mmol) was stirred at 120 °C for 14 h. The reaction mixture was cooled, poured into water and extracted with CH_2_Cl_2_. The organic layer was separated, washed with water, dried over anhydrous MgSO_4_ and filtered. The solvent was removed at reduced pressure, and the residue was purified using silica gel column chromatography (eluent petroleum ether—EtOAc, 25:1) to give the title product. Yield (260 mg, 65%) colorless oil, *R_f_* 0.60 (1:0.3 hexane/EtOAc). IR (KBr, *ν*_max_/cm^−1^): 2936 (CH), 2857 (CH), 1736 (C=O), 1594 (C-N, C=C), 1571, 1459, 1365, 1334, 1285, 1241, 1128, 1106, 1039, 929, 801. ^1^H NMR (400 MHz, CDCl_3_) *δ* 7.17–7.13 (m, 4H), 6.93–6.89 (m, 2H), 6.87–6.84 (m, 2H), 4.04 (t, *J* = 6.7 Hz, 2H, CH_2_OC(O)CH_3_), 3.85 (t, *J* = 7.1 Hz, 2H, NCH_2_), 2.03 (s, 3H, CH_3_), 1.87–1.78 (m, 2H, NCH_2_CH_2_(CH_2_)_2_CH_2_CH_2_OC(O)CH_3_), 1.65–1.58 (m, 2H, NCH_2_CH_2_(CH_2_)_2_CH_2_CH_2_OC(O)CH_3_), 1.50–1.33 (m, 4H, NCH_2_CH_2_(CH_2_)_2_ CH_2_CH_2_OC(O)CH_3_). ^13^C NMR (100 MHz, CDCl_3_) *δ* 171.0 (C), 145.2 (C), 127.4 (CH), 127.1 (CH), 125.0 (C), 122.3 (CH), 115.3 (CH), 64.3 (CH), 47.1 (CH), 28.5 (CH), 26.7 (CH), 26.5 (CH), 25.5 (CH), 20.9 (CH).

### 3.4. 6-(3-Formyl-10H-phenothiazin-10-yl)hexyl Acetate (**1e**)

To a mixture of 6-(10H-phenothiazin-10-yl)hexyl acetate (230 mg, 0.67 mmol), anhydrous DMF (86 mg, 1.18 mmol) and 1,2-dichloroethane (1 mL), POCl_3_ (181 mg, 1.18 mmol) were added dropwise at 0 °C. The reaction mixture was refluxed for 9 h, cooled, poured into water and extracted with CH_2_Cl_2_. The organic layer was separated, washed with water, dried over anhydrous MgSO_4_ and filtered. The solvent was removed at reduced pressure, and the residue was purified using silica gel column chromatography (eluent petroleum ether—EtOAc, gradient from 50:1 to 10:1) to give **1e**. Yield (190 mg, 77%) yellow oil, *R_f_* 0.27 (1:0.3 hexane/EtOAc). ^1^H NMR (400 MHz, CDCl_3_) *δ* 9.77 (s, 1H, C(O)H), 7.61 (dd, *J* = 8.4, 1.9 Hz, 1H), 7.55 (d, *J* = 1.9 Hz, 1H), 7.15 (ddd, *J* = 8.1, 7.3, 1.6 Hz, 1H), 7.09 (dd, *J* = 7.6, 1.6 Hz, 1H), 6.94 (td, *J* = 7.5, 1.2 Hz, 1H), 6.88–6.84 (m, 2H), 4.01 (t, *J* = 6.7 Hz, 2H, CH_2_OC(O)CH_3_), 3.87 (t, *J* = 7.1 Hz, 2H, NCH_2_), 2.00 (s, 3H, CH_3_), 1.84–1.76 (m, 2H, NCH_2_CH_2_(CH_2_)_2_CH_2_CH_2_OC(O)CH_3_), 1.63–1.56 (m, 2H, NCH_2_CH_2_(CH_2_)_2_CH_2_CH_2_OC(O)CH_3_), 1.49–1.32 (m, 4H, NCH_2_CH_2_(CH_2_)_2_ CH_2_CH_2_OC(O)CH_3_). ^13^C NMR (100 MHz, CDCl_3_) *δ* 189.9 (CH), 171.1 (C), 150.7 (C), 143.4 (C), 131.1 (C), 130.0 (CH), 128.4 (CH), 127.60 (CH), 127.57 (CH), 125.2 (C), 124.0 (C), 123.6 (CH), 116.0 (CH), 114.9 (CH), 64.3 (CH), 47.8 (CH), 28.5 (CH), 26.7 (CH), 26.4 (CH), 25.6 (CH), 21.0 (CH). 

### 3.5. General Procedure for Synthesis of Compounds **2**

To a mixture of methyltriphenylphosphonium bromide, THF and t-BuOK were added under continuous stirring, and the resulting mixture was kept at 0 °C for 30 min. Then, a solution of aldehyde **1** in 2 mL of THF was added dropwise to the obtained mixture, and the mixture was stirred at room temperature for 2–5 h. The THF was removed in vacuum, and the residue was purified using column chromatography on silica gel (eluent petroleum ether/EtOAc, gradient from 10:1 to 4:1) to give **2**.

#### 3.5.1. ((4-Vinylphenyl)azanediyl)bis(ethane-2,1-diyl) Diacetate (**2b**)

The use of methyltriphenylphosphonium bromide (1.50 g, 4.20 mmol), *t*-BuOK (0.70 g, 6.00 mmol), THF (3 mL) and compound **1b** (1.00 g, 3.41 mmol) in general procedure afforded the title compound **2b** (0.57 g, 57%) as colorless oil, *R_f_* 0.55 (4:1 hexane/EtOAc). IR (KBr, *ν*_max_/cm^−1^): 2963 (CH), 2923 (CH), 2855 (CH), 1740 (C=O), 1625, 1610, 1559, 1520, 1450, 1439, 1383, 1333, 1238, 1190, 1069, 1052, 995, 828. ^1^H NMR (400 MHz, CDCl_3_) *δ* 7.30 (d, *J* = 8.8 Hz, 2H, H-3,5-aniline), 6.71 (d, *J* = 8.8 Hz, 2H, H-2,6-aniline), 6.61 (dd, *J* = 17.5. 10.9 Hz, 1H, -HC=CH_2_), 5.54 (d, *J* = 16.5 Hz, 1H, -HC=CH_2_), 5.03 (d, *J* = 10.3 Hz, 1H, -HC=CH-), 4.24 (t, *J* = 6.3 Hz, 2H, OCH_2_), 3.62 (t, *J* = 6.3 Hz, 2H, NCH_2_), 2.04 (s, 6H, CH_3_). ^13^C NMR (125 MHz, CDCl_3_) *δ* 170.8 (C), 146.8 (C), 136.2 (CH), 127.4 (CH), 126.7 (C), 111.9 (CH), 109.7 (CH), 61.3 (CH), 49.6 (CH), 20.7 (CH).

#### 3.5.2. 6-(Ethyl(4-vinylphenyl)amino)hexyl Acetate (**2c**)

The use of methyltriphenylphosphonium bromide (1.47 g, 4.1 mmol), t-BuOK (0.46 g, 4.1 mmol), THF (3 mL) and compound **1c** (1.00 g, 3.43 mmol) in general procedure afforded the title compound **2c** (484 mg, 51%) as colorless oil, *R_f_* 0.38 (10:3 hexane/EtOAc). IR (KBr, *ν*_max_/cm^−1^): 2965 (CH), 2922 (CH), 2857 (CH), 1739 (C=O), 1623, 1611, 1559, 1522, 1451, 1439, 1383, 1331, 1238, 1190, 1069, 1052, 995, 828. ^1^H NMR (400 MHz, CDCl_3_) *δ* 7.33 (d, *J* = 8.8 Hz, 2H, *o*-Ph), 6.68–6.64 (m, 3H, *m*-Ph, -CH=CH_2_), 5.56 (dd, *J* = 17.6, 1.2 Hz, 1H), 5.04 (dd, *J* = 10.9, 1.2 Hz, 1H), 4.12 (t, *J* = 6.7 Hz, 2H, NCH_2_(CH_2_)_4_CH_2_OC(O)CH_3_), 3.41 (q, *J* = 7.0 Hz, 2H, NCH_2_(CH_2_)_4_CH_2_OC(O)CH_3_), 3.31 (t, *J* = 7.6 Hz, 2H, NCH_2_CH_3_), 2.10 (s, 3H, CH_3_), 1.73–1.62 (m, 4H), 1.50–1.38 (m, 4H), 1.20 (t, *J* = 7.1 Hz, 3H, CH_3_). ^13^C NMR (100 MHz, CDCl_3_) *δ* 170.9 (C), 147.5 (C), 136.5 (CH), 127.2 (CH), 125.1 (C), 111.5 (CH), 108.5 (CH), 64.3 (CH), 50.2 (CH), 44.9 (CH), 28.5 (CH), 27.4 (CH), 26.7 (CH), 25.8 (CH), 20.8 (CH), 12.2 (CH).

#### 3.5.3. 6-(3-Vinyl-10H-phenothiazin-10-yl)hexan-1-ol (**2e′**)

The use of methyltriphenylphosphonium bromide (304 mg, 0.85 mmol), t-BuOK (159 mg, 1.42 mmol), THF (3 mL) and compound **1e** (262 mg, 0.71 mmol) in general procedure afforded the title compound **2e′** (114 mg, 50%) as colorless oil, *R_f_* 0.18 (10:3 hexane/EtOAc). IR (KBr, *ν*_max_/cm^−1^): 3431 (OH), 2925 (CH), 2854 (CH), 1627, 1599, 1575, 1464, 1376, 1332, 1247, 1161, 1075, 1039, 987, 816. ^1^H NMR (600 MHz, CDCl_3_) *δ* 7.21–7.13 (m, 4H), 6.91 (t, *J* = 7.5 Hz, 1H), 6.84 (d, *J* = 8.1 Hz, 1H), 6.79 (d, *J* = 8.3 Hz, 1H), 6.62–6.57 (m, 1H, CH=CH_2_), 5.62 (d, *J* = 17.5 Hz, 1H, CH=CH_2_), 5.14 (d, *J* = 10.8 Hz, 1H, CH=CH_2_), 3.84 (t, *J* = 7.1 Hz, 2H, NCH_2_), 3.59 (t, *J* = 6.5 Hz, 2H, CH_2_OH), 1.83–1.78 (m, 2H, NCH_2_CH_2_(CH_2_)_2_CH_2_CH_2_OH), 1.58 (s, 1H, CH_2_OH), 1.57–1.52 (m, 2H, NCH_2_CH_2_(CH_2_)_2_CH_2_CH_2_OH), 1.48–1.43 (m, 2H, N(CH_2_)_2_CH_2_CH_2_(CH_2_)_2_OH), 1.40–1.35 (m, 2H, N(CH_2_)_2_CH_2_CH_2_(CH_2_)_2_OH). ^13^C NMR (150 MHz, CDCl_3_) *δ* 144.9 (C), 144.7 (C), 135.5 (CH), 132.1 (C), 127.4 (CH), 127.2 (CH), 125.4 (CH), 125.0 (C), 124.8 (CH), 124.5 (C), 122.3 (CH), 115.3 (CH), 115.2 (CH), 112.1 (CH), 62.6 (CH), 47.2 (CH), 32.5 (CH), 26.7 (CH), 26.6 (CH), 25.3 (CH). 

#### 3.5.4. 1-(6-((tert-Butyldimethylsilyl)oxy)hexyl)-6-vinyl-1,2,3,4-tetrahydroquinoline (**2f′**)

The use of methyltriphenylphosphonium bromide (317 mg, 0.89 mmol), *t*-BuOK (111 mg, 0.94 mmol), **1f′** (190 mg, 0.51 mmol) and THF (9.0 mL) in general procedure afforded the title compound **2f’** (140 mg, 73%) as colorless oil, *R_f_* 0.75 (10:1 hexane/EtOAc. IR (*ν*_max_, cm^−1^, KBr): 2931 (CH), 2857 (CH), 1609 (C=C), 1513, 1463, 1407, 1343, 1309, 1254, 1200, 1102, 988, 939, 883, 836, 807, 776. ^1^H NMR (500 MHz, CDCl_3_): *δ* 7.11 (dd, *J* = 8.5, 2.0 Hz, 1H, H-7 quinoline), 7.04 (d, *J* = 2.0 Hz, 1H, H-5 quinoline), 6.59 (dd, *J* = 17.5 Hz, 10.8 Hz, 1H, -CH=CH_2_), 6.52 (d, *J* = 8.5 Hz, 1H, H-8 quinoline), 5.47 (dd, *J* = 17.5 Hz, 1.0 Hz, 1H, CH=CH_2_), 4.95 (dd, *J* = 10.9 Hz, 1.0 Hz, 1H, CH=CH_2_), 3.63 (t, *J* = 6.4 Hz, 2H, OCH_2_), 3.29 (t, *J* = 5.6 Hz, 2H, H-2 tertahydroquinoline), 3.26 (t, *J* = 7.5 Hz, 2H, NCH_2_), 2.76 (t, *J* = 6.3 Hz, 2H, H-4 quinoline), 1.99–1.91 (m, 2H, H-3 quinoline), 1.65–1.52 (m, 4H, CH_2_), 1.43–1.34 (m, 4H, CH_2_), 0.92 (s, 9H, CH_3_), 0.08 (s, 6H, CH_3_). ^1^H NMR (400 MHz, CDCl_3_): δ 145.1 (C), 136.8 (CH), 126.9 (CH), 125.4 (CH), 125.0 (C), 121.9 (C), 110.2 (CH), 108.3 (CH), 63.2 (CH), 51.4 (CH), 49.5 (CH), 32.9 (CH), 28.2 (CH), 27.0 (CH), 26.2 (CH), 26.0 (CH), 25.7 (CH), 22.2 (CH), 18.4 (C), −5.44 (CH).

### 3.6. General Procedure for Synthesis of **4a–d, 5e′** and **7f**

A mixture of 6-bromoquinoxaline **3** (**3′**), 4-vinylaniline **2**, tri(*o*-tolyl)phosphine, Pd(OAc)_2_, Et_3_N and anhydrous DMF was stirred at 120 °C for 3–10 h. The reaction mixture was cooled, poured into water and extracted with CH_2_Cl_2_. The organic layer was separated, washed with water, dried over anhydrous MgSO_4_ and filtered. The solvent was removed at reduced pressure, and the residue was purified using silica gel column chromatography (eluent petroleum ether/EtOAc, gradient from 25:1 to 4:1) to give **4a–d**, **5e′** and **7f**.

#### 3.6.1. (E)-((4-(2-(2-Methyl-3-phenylquinoxalin-6-yl)vinyl)phenyl)azanediyl)bis(ethane-2,1-diyl) diacetate (**4b**)

The use of **3** (388 mg, 1.3 mmol), **2b** (378 mg, 1.3 mmol), tri(*o*-tolyl)phosphine (40.0 mg, 0.13 mmol), Pd(OAc)_2_ (15 mg, 0.062 mmol) and Et_3_N (330 mg, 3.3 mmol) (reaction time 3 h) in general procedure afforded the title compound **4b** (470 mg, 71%) as orange oil, *R_f_* 0.14 (1:1 hexane/EtOAc). IR (KBr, *ν*_max_/cm^−1^): 3025 (CH), 2963 (CH), 1738 (C=O), 1599, 1519, 1395, 1349, 1227, 1185, 1020, 897. ^1^H NMR (500 MHz, CDCl_3_) *δ* 8.06 (d, *J* = 1.7 Hz, 1H, H-5 quinoxaline), 7.97 (d, *J* = 8.8 Hz, 1H, H-8 quinoxaline), 7.94 (dd, *J* = 8.8, 1.7 Hz, 1H, H-7 quinoxaline), 7.67–7.63 (m, 2H, *o*-Ph), 7.55–7.47 (m, 3H, *m,p*-Ph), 7.45 (d, *J* = 8.9 Hz, 2H, H-3,5-aniline), 7.22 (d, *J* = 16.2 Hz, 1H, -HC=CH-), 7.08 (d, *J* = 16.2 Hz, 1H, -HC=CH-), 6.77 (d, *J* = 8.9 Hz, 2H, H-2,6-aniline), 4.26 (t, *J* = 6.3 Hz, 4H, OCH_2_), 3.65 (t, *J* = 6.3 Hz, 4H, NCH_2_), 2.75 (s, 3H, CH_3_), 2.05 (s, 6H, CH_3_). ^13^C NMR (125 MHz, CDCl_3_) *δ* 170.8 (C), 155.0 (C), 151.4 (C), 147.3 (C), 141.5 (C), 140.7 (C), 139.1 (C), 130.8 (CH), 128.9 (CH), 128.8 (CH), 128.5 (CH), 128.3 (CH), 128.2 (CH), 127.8 (CH), 125.9 (C), 125.7 (CH), 123.7 (CH), 112.1 (CH), 61.3 (CH), 49.7 (CH), 24.2 (CH), 20.8 (CH). HRMS (ESI) calcd for C_31_H_32_N_3_O_4_ [M+H]^+^ 510.2387, found 510.2389.

#### 3.6.2. (E)-6-(Ethyl(4-(2-(2-methyl-3-phenylquinoxalin-6-yl)vinyl)phenyl)amino)hexyl Acetate (**4c**)

The use of **3** (370 mg, 1.24 mmol), **2c** (358 mg, 1.24 mmol), tri(*o*-tolyl)phosphine (38 mg, 0.13 mmol), Pd(OAc)_2_ (14 mg, 0.062 mmol) and Et_3_N (313 mg, 3.1 mmol) (reaction time 8 h) in general procedure afforded the title compound **3c** (372 mg, 59%) as orange oil, *R_f_* 0.33 (10:3 hexane/EtOAc). IR (KBr, *ν*_max_/cm^−1^): 2933 (CH), 1735 (C=O), 1599, 1521, 1349, 1244, 1186, 1005, 829. ^1^H NMR (400 MHz, CDCl_3_) *δ* 8.05 (d, *J* = 1.7 Hz, 1H, H-5 quinoxaline), 7.97 (d, *J* = 8.8 Hz, 1H, H-8 quinoxaline), 7.93 (dd, *J* = 8.8, 1.7 Hz, 1H, H-7 quinoxaline), 7.68–7.64 (m, 2H, *o*-Ph), 7.56–7.47 (m, 3H, *m,p*-Ph), 7.43 (d, *J* = 8.8 Hz, 2H, H-3,5-aniline), 7.23 (d, *J* = 16.2 Hz, 1H), 7.05 (d, *J* = 16.2 Hz, 1H), 6.66 (d, *J* = 8.8 Hz, 2H, H-2,6-aniline), 4.07 (t, *J* = 6.7 Hz, 2H, OCH_2_), 3.39 (q, *J* = 7.0 Hz, 2H, NCH_2_), 3.29 (t, *J* = 7.6 Hz, 2H, NCH_2_), 2.75 (s, 3H), 2.05 (s, 3H), 1.67–1.58 (m, 4H), 1.45–1.35 (m, 4H), 1.17 (t, *J* = 7.0 Hz, 3H). ^13^C NMR (100 MHz, CDCl_3_) *δ* 171.1 (C), 155.0 (C), 151.2 (C), 148.0 (C), 141.6 (C), 140.6 (C), 139.5 (C), 139.2 (C), 131.3 (CH), 129.0 (CH), 128.8 (CH), 128.5 (CH), 128.2 (2CH), 127.8 (CH), 125.4 (CH), 124.2 (C), 122.5 (CH), 111.7 (CH), 64.4 (CH), 50.3 (CH), 45.0 (CH), 28.6 (CH), 27.5 (CH), 26.8 (CH), 25.8 (CH), 24.2 (CH), 20.9 (CH), 12.4 (CH). HRMS (ESI) calcd for C_33_H_38_N_3_O_2_ [M+H]^+^ 508.2958, found 508.2965.

#### 3.6.3. (E)-6-(3-(2-(2-Methyl-3-phenylquinoxalin-6-yl)vinyl)-10H-phenothiazin-10-yl)hexan-1-ol (**5e**)

The use of **3** (59 mg, 0.2 mmol), **2e′** (64 mg, 0.2 mmol), tri(*o*-tolyl)phosphine (6 mg, 0.02 mmol), Pd(OAc)_2_ (2 mg, 0.01 mmol) and Et_3_N (50 mg, 0.5 mmol) (reaction time 8 h) in general procedure afforded the title compound **5e** (80 mg, 75%) as orange powder. Mp 77–78 °C, *R_f_* 0.29 (10:2 hexane/EtOAc). IR (KBr, *ν*_max_/cm^−1^): 3422 (OH), 2926 (CH), 2853 (CH), 1597, 1574, 1465, 1347, 1245, 1106, 1005, 827. ^1^H NMR (400 MHz, CDCl_3_) *δ* 8.09 (d, *J* = 1.7 Hz, 1H, H-5 quinoxaline), 8.00 (d, *J* = 8.8 Hz, 1H, H-8 quinoxaline), 7.92 (dd, *J* = 8.8, 1.7 Hz, 1H, H-7 quinoxaline), 7.69–7.64 (m, 2H, *o*-Ph), 7.56–7.47 (m, 3H, *m,p*-Ph), 7.33 (d, *J* = 8.4, 1.8 Hz, 1H, H-2 phenothiazine), 7.31 (d, *J* = 1.8 Hz, 1H, H-4 phenothiazine), 7.18–7.10 (m, 4H), 6.95–6.89 (m, 1H, H7 or H8 phenothiazine), 6.87–6.81 (m, 2H, phenothiazine), 3.86 (t, *J* = 7.1 Hz, 2H, NCH_2_), 3.61 (t, *J* = 6.5 Hz, 2H, OCH_2_), 2.76 (s, 3H), 1.87–1.76 (m, 2H), 1.60–1.51 (m, 2H), 1.50–1.33 (m, 4H). ^13^C NMR (100 MHz, CDCl_3_) *δ* 155.2 (C), 151.8 (C), 145.1 (C), 144.7 (C), 141.5 (C), 140.8 (C), 139.1 (C), 138.6 (C), 131.4 (C), 129.8 (CH), 129.0 (CH), 128.9 (CH + C), 128.5 (CH), 128.4 (CH), 127.8 (CH), 127.5 (CH), 127.3 (CH), 126.3 (CH), 126.2 (CH), 125.9 (CH), 125.3 (CH), 124.3 (C), 122.6 (CH), 115.43 (CH), 115.36 (CH), 62.7 (CH), 47.4 (CH), 32.6 (CH), 26.8 (CH), 26.7 (CH), 25.4 (CH), 24.2 (CH). HRMS (ESI) calcd for C_35_H_34_N_3_OS [M+H]^+^ 544.2417, found 544.2412.

#### 3.6.4. (E)-6-(2-(1-(6-((tert-butyldimethylsilyl)oxy)hexyl)-1,2,3,4-tetrahydroquinolin-6-yl)vinyl)-3-phenylquinoxaline-2-carbaldehyde (**7f**)

The use of **3′** (44.6 mg, 0.14 mmol), **2f′** (56 mg, 0.15 mmol), tri(*o*-tolyl)phosphine (4.4 mg, 0.014 mmol), Pd(OAc)_2_ (1.6 mg, 0.007 mmol) and Et_3_N (36 mg, 0.36 mmol) (reaction time 10 h) in general procedure afforded the title compound **7f** (11 mg, 14 %) as a red oil, *R_f_* 0.26 (9:1 hexane/EtOAc). IR (KBr, ν, cm^–1^): 2930 (CH), 2856 (CH), 1716 (C=O), 1595 (C=N, C=C), 1514, 1485, 1426, 1346, 1252, 1197, 1152, 1102, 1024, 956, 888, 836, 805, 776. ^1^H NMR (500 MHz, CDCl_3_): *δ* 10.27 (s, 1H, C(O)H), 8.20 (d, *J* = 8.8 Hz, 1H, H-5 quinoxaline), 8.07 (d, *J* = 2.0 Hz, 1H, H-8 quinoxaline), 8.05 (dd, *J* = 8.8 Hz, 2.0 Hz, 1H, H-6 quinoxaline), 7.72–7.69 (m, 2H, *o*-Ph), 7.57–7.54 (m, 3H, *m, p*-Ph), 7.30 (d, *J* = 16.2 Hz, 1H, -CH=CH-), 7.28 (dd, *J* = 8.4 Hz, 2.0 Hz, 1H, H-7 quinoline), 7.21 (d, *J* = 2.0 Hz, 1H, H-5 quinoline), 7.04 (d, *J* = 16.2 Hz, 1H, -CH=CH-), 6.56 (d, *J* = 8.4 Hz, 1H, H-8 quinoline), 3.62 (t, *J* = 6.5 Hz, 2H, OCH_2_), 3.34 (t, J = 5.6 Hz, 2H, H-2 quinoline), 3.29 (t, *J* = 7.5 Hz, 2H, NCH_2_), 2.80 (t, *J* = 6.2 Hz, 2H, H-4 quinoline), 2.01–1.92 (m, H-3 quinoline), 1.65–1.60 (m, 2H, CH_2_), 1.56–1.52 (m, 2H, CH_2_), 1.40–1.35 (m, 4H, CH_2_), 0.90 (s, 9H, CH_3_), 0.06 (s, 6H, CH_3_). ^1^H NMR (125 MHz, CDCl_3_) *δ* 191.1 (CH), 155.2 (C), 146.2 (C), 143.9 (C), 143.8 (C), 143.4 (C), 140.7 (C), 136.9 (C), 134.1 (CH), 130.2 (CH), 129.8 (CH), 129.7 (CH), 129.2 (CH), 128.6 (CH), 127.9 (CH), 127.2 (CH), 124.7 (CH), 123.5 (C), 122.3 (C), 121.4 (CH), 110.3 (CH), 63.1 (CH), 51.4 (CH), 49.6 (CH), 32.8 (CH), 28.3 (CH), 27.0 (CH), 26.4 (CH), 26.0 (CH), 25.8 (CH), 22.0 (CH), 18.4 (C), −5.2 (CH). 

### 3.7. General Procedure for Synthesis of **5a–d**

A mixture of compound **4** in CH_2_Cl_2_, EtOH or MeOH, and a 50% aqueous solution of KOH or 10% aqueous solution of NaOH was stirred at room temperature for 15 min–18 h. The mixture was neutralized with several drops of acetic acid and washed with water. The product was extracted with CH_2_Cl_2_, and the organic layer was dried over MgSO_4_ and filtered. The solvent was removed under reduced pressure. The product was purified using column chromatography on silica gel (elution CH_2_Cl_2_–CH_3_OH, gradient from 150:1 to 5:1 or eluent petroleum ether—EtOAc, gradient from 10:1 to 1:1) to give **5a–d**.

#### 3.7.1. (E)-2,2′-((4-(2-(2-Methyl-3-phenylquinoxalin-6-yl)vinyl)phenyl)azanediyl)bis(ethan-1-ol) (**5b**)

The use of compound **4b** (245 mg, 0.48 mmol), CH_2_Cl_2_ (3 mL), EtOH (3 mL) and a 50% aqueous solution of KOH (5 mL) (reaction time 18 h) in general procedure afforded the title compound **5b** (147 mg, 72%) as orange powder. Mp 86–88 °C (hexane), *R_f_* 0.28 (EtOAc). IR (KBr, *ν*_max_/cm^−1^): 3504 (OH), 2962 (CH), 2924 (CH), 2853 (CH), 2833 (CH), 1603, 1516, 1446, 1377, 1344, 1232, 1190, 1120, 1049, 1005, 926, 835, 708. ^1^H NMR (400 MHz, CDCl_3_) δ 8.07 (d, *J* = 1.7 Hz, 1H, H-5 quinoxaline), 7.98 (d, *J* = 8.8 Hz, 1H, H-8 quinoxaline), 7.94 (dd, *J* = 8.8, 1.7 Hz, 1H, H-7 quinoxaline), 7.69–7.64 (m, 2H, *o*-Ph), 7.56–7.48 (m, 3H, *m,p*-Ph), 7.46 (d, *J* = 8.8 Hz, 2H, H-3,5-aniline), 7.24 (d, *J* = 16.1 Hz, 1H, -HC=CH-), 7.09 (d, *J* = 16.1 Hz, 1H, -HC=CH-), 6.73 (d, *J* = 8.8 Hz, 2H, H-2,6-aniline), 3.92 (t, *J* = 4.9 Hz, 4H, OCH_2_), 3.65 (t, *J* = 4.9 Hz, 4H, NCH_2_), 2.83 (br, 2H, OH), 2.76 (s, 3H, CH_3_). ^13^C NMR (125 MHz, CDCl_3_) *δ* 155.1 (C), 151.4 (C), 147.9 (C), 141.6 (C), 140.7 (C), 139.3 (C), 139.2 (C), 131.0 (CH), 129.0 (2CH), 128.5 (CH), 128.19 (CH), 128.16 (CH), 127.9 (CH), 125.8 (C), 125.7 (CH), 123.6 (CH), 112.6 (CH), 60.8 (CH), 55.2 (CH), 24.2 (CH). HRMS (ESI) calcd for C_27_H_28_N_3_O_2_ [M+H]^+^ 426.2176, found 426.2176. 

#### 3.7.2. (E)-6-(Ethyl(4-(2-(2-methyl-3-phenylquinoxalin-6-yl)vinyl)phenyl)amino)hexan-1-ol (**5c**)

The use of compound **4c** (65 mg, 1.28 mmol), CH_2_Cl_2_ (0.5 mL), MeOH (0.5 mL) and a 10% aqueous solution of NaOH (1.3 mL) (reaction time 1.5 h) in general procedure afforded the title compound **5c** (49 mg, 82%) as orange oil, *R_f_* 0.11 (10:3 hexane/EtOAc). IR (KBr, *ν*_max_ /cm^−1^): 3473 (OH), 2970 (CH), 2929 (CH), 2868 (CH), 1679, 1599 (C-N, C=C), 1521, 1452, 1373, 1348, 1297, 1273, 1167, 1112, 1007, 930, 830. ^1^H NMR (400 MHz, CDCl_3_) *δ* 8.05 (s, 1H, H-5 quinoxaline), 7.98 (d, *J* = 8.9 Hz, 1H, H-8 quinoxaline), 7.95 (dd, *J* = 8.9, 1.7 Hz, 1H, H-7 quinoxaline), 7.68–7.63 (m, 2H, *o*-Ph), 7.55–7.47 (m, 3H, *m,p*-Ph), 7.43 (d, *J* = 8.6 Hz, 2H, H-3,5-aniline), 7.23 (d, *J* = 16.1 Hz, 1H), 7.05 (d, *J* = 16.1 Hz, 1H), 6.66 (d, *J* = 8.6 Hz, 2H), 3.65 (t, *J* = 6.5 Hz, 2H, OCH_2_), 3.40 (q, *J* = 7.0 Hz, 2H, NCH_2_), 3.29 (t, *J* = 7.0 Hz, 2H, NCH_2_), 2.75 (s, 3H, CH_3_), 1.66–1.55 (m, 4H, NCH_2_(CH_2_)_2_(CH_2_)_2_ or NCH_2_(CH_2_)_2_, 1.45–1.38 (m, 4H, NCH_2_(CH_2_)_2_ or NCH_2_(CH_2_)_2_(CH_2_)_2_), 1.17 (t, *J* = 7.0 Hz, 3H, CH_3_). ^13^C NMR (100 MHz, CDCl_3_) *δ* 155.0 (C), 151.2 (C), 148.0 (C), 141.6 (C), 140.6 (C), 139.6 (C), 139.3 (C), 131.4 (M+H), 129.0 (CH), 128.9 (CH), 128.5 (CH), 128.23 (CH), 128.19 (CH), 127.9 (CH), 125.4 (CH), 124.1 (C), 122.5 (CH), 111.7 (CH), 62.8 (CH), 50.4 (CH), 45.0 (CH), 32.8 (CH), 27.6 (CH), 27.0 (CH), 25.7 (CH), 24.2 (CH), 12.4 (CH). HRMS (ESI) calcd for C_31_H_36_N_3_O [M+H]^+^ 466.2853, found 466.2851. 

### 3.8. General Procedure for Synthesis of **6a–e** and **1f**′

A mixture of compound **5** or 1-(6-hydroxyhexyl)-1,2,3,4-tetrahydroquinoline-6-carbaldehyde, imidazole, *tert*-butyldimethylsilyl chloride and DMF was stirred at 50 °C for 2–5 h. The mixture was cooled, poured into water and extracted with CH_2_Cl_2_. The organic layer was dried over MgSO_4_ and filtered; the solvent was evaporated. The residue was purified using column chromatography on silica gel (eluent petroleum ether/EtOAc, gradient from 100:1 to 10:1) to give **6a–e** and **1f’**.

#### 3.8.1. (E)-N,N-Bis(2-((tert-butyldimethylsilyl)oxy)ethyl)-4-(2-(2-methyl-3-phenylquinoxalin-6-yl)vinyl)aniline (**6b**)

The use of compound **5b** (100 mg, 0.24 mmol), imidazole (32 mg, 0.48 mmol), *tert*-butyldimethylsilyl chloride (46 mg, 0.31 mmol) and DMF (0.5 mL) (reaction time 4 h) in general procedure afforded the title compound **6b** (100 mg, 65%) as yellow powder. Mp 98–100 °C, *R_f_* 0.32 (10:3 hexane/EtOAc). IR (KBr, *ν*_max_/cm^−1^): 2954 (CH), 2929 (CH), 2885 (CH), 2857 (CH), 1600 (C=C, C=N), 1522, 1489, 1472, 1425, 1392, 1350, 1304, 1292, 1273, 1253, 1227, 1184, 1132, 1091, 1072, 997, 960, 914, 830, 773. ^1^H NMR (400 MHz, CDCl_3_) *δ* 8.06 (s, 1 H, H-5 quinoxaline), 7.98 (d, *J* = 8.9 Hz, 1H, H-8 quinoxaline), 7.95 (dd, *J* = 8.9, 2.1 Hz, 1H, H-7 quinoxaline), 7.66–7.53 (d, *J* = 7.5 Hz, 2H, *o*-Ph), 7.55–7.47 (m, 3H, *m,p*-Ph), 7.42 (d, *J* = 8.7 Hz, 2H, H-3,5-aniline), 7.23 (d, *J* = 16.2 Hz, 1H, -HC=CH-), 7.07 (d, *J* = 16.2 Hz, 1H, -HC=CH-), 6.70 (d, *J* = 8.7 Hz, 2H, H-2,6-aniline), 3.78 (t, *J* = 6.3 Hz, 4H, OCH_2_), 3.56 (t, *J* = 6.3 Hz, 4H, NCH_2_), 2.78 (s, 3H, CH_3_), 0.90 (s, 18H, CH_3_), 0.05 (s, 12H, CH_3_). ^13^C NMR (100 MHz, CDCl_3_) *δ* 155.0 (C), 151.3 (C), 148.1 (C), 141.7 (C), 140.7 (C), 139.5 (C), 139.3 (C), 131.3 (CH), 129.0 (CH), 128.9 (CH), 128.5 (CH), 128.3 (CH), 128.2 (CH), 127.9 (CH), 125.5 (CH), 124.6 (C), 122.8 (CH), 111.7 (CH), 60.3 (CH), 53.6 (CH), 25.9 (CH), 24.2 (CH), 18.3 (C), −5.3 (CH). HRMS (ESI) calcd for C_39_H_56_N_3_O_2_Si_2_ [M+H]^+^ 654.3905, found 654.3900. 

#### 3.8.2. (E)-N-(6-((tert-Butyldimethylsilyl)oxy)hexyl)-N-ethyl-4-(2-(2-methyl-3-phenylquinoxalin-6-yl)vinyl)aniline (**6c**)

The use of compound **5c** (49 mg, 0.10 mmol), imidazole (14 mg, 0.2 mmol), *tert*-butyldimethylsilyl chloride (21 mg, 0.13 mmol) and DMF (0.5 mL) (reaction time 2 h) in general procedure afforded the title compound **6c** (50 mg, 82%) as orange oil, *R_f_* 0.37 (5:1 hexane/EtOAc). IR (KBr, *ν*_max_/cm^−1^): 2926 (CH), 2855 (CH), 1600 (C-N, C=C), 1520, 1426, 1372, 1349, 1252, 1186, 1099, 1005, 956, 830. ^1^H NMR (400 MHz, CDCl_3_) *δ* 8.05 (d, *J* = 1.7 Hz, 1H, H-5 quinoxaline), 7.98 (d, *J* = 8.8 Hz, 1H, H-8 quinoxaline), 7.94 (dd, *J* = 8.8, 1.7 Hz, 1H, H-7 quinoxaline), 7.69–7.64 (m, 2H, *o*-Ph), 7.56–7.47 (m, 3H, *m,p*-Ph), 7.44 (d, *J* = 8.8 Hz, 2H, H-3,5-aniline), 7.24 (d, *J* = 16.2 Hz, 1H), 7.05 (d, *J* = 16.2 Hz, 1H), 6.66 (d, *J* = 8.8 Hz, 2H, H-2,6-aniline), 3.63 (t, *J* = 6.5 Hz, 2H, OCH_2_), 3.40 (q, *J* = 7.0 Hz, 2H, NCH_2_), 3.29 (t, *J* = 7.6 Hz, 2H, NCH_2_), 2.76 (s, 3H), 1.68–1.59 (m, 4H), 1.45–1.34 (m, 4H), 1.18 (t, *J* = 7.0 Hz, 3H), 0.92 (s, 9H, CH_3_), 0.07 (s, 6H, CH_3_). ^13^C NMR (100 MHz, CDCl_3_) *δ* 155.0 (C), 151.2 (C), 148.0 (C), 141.6 (C), 140.6 (C), 139.6 (C), 139.3 (C), 131.4 (CH), 128.9 (CH), 128.8 (CH), 128.5 (CH), 128.21 (CH), 128.19 (CH), 127.8 (CH), 125.4 (CH), 124.1 (C), 122.5 (CH), 111.6 (CH), 63.1 (CH), 50.4 (CH), 45.0 (CH), 32.8 (CH), 27.6 (CH), 27.0 (CH), 26.0 (CH), 25.8 (CH), 24.2 (CH), 18.4 (C), 12.4 (CH), −5.3 (CH). HRMS (ESI) calcd for C_37_H_50_N_3_OSi [M+H]^+^ 580.3718, found 580.3729.

#### 3.8.3. (E)-10-(6-((tert-Butyldimethylsilyl)oxy)hexyl)-3-(2-(2-methyl-3-phenylquinoxalin-6-yl)vinyl)-10H-phenothiazine (**6e**)

The use of compound **5e** (20 mg, 0.04 mmol), imidazole (5 mg, 0.1 mmol), *tert*-butyldimethylsilyl chloride (7 mg, 0.05 mmol) and DMF (0.5 mL) (reaction time 5 h) in general procedure afforded the title compound **6e** (19 mg, 79%) as orange powder. Mp 45–46 °C, *R_f_* 0.38 (5:1 hexane/EtOAc). IR (KBr, *ν*_max_/cm^−1^): 2927 (CH), 2855 (CH), 1600 (C-N, C=C), 1466, 1347, 1245, 1106, 1005, 827. ^1^H NMR (400 MHz, CDCl_3_) *δ* 8.09 (d, *J* = 1.8 Hz, 1H, H-5 quinoxaline), 8.00 (d, *J* = 8.8 Hz, 1H, H-8 quinoxaline), 7.93 (dd, *J* = 8.8, 1.8 Hz, 1H, H-7 quinoxaline), 7.68–7.64 (m, 2H, *o*-Ph), 7.56–7.48 (m, 3H, *m,p*-Ph), 7.34 (d, *J* = 1.9 Hz, 1H, H-4 phenothiazine), 7.31 (dd, *J* = 8.4, 1.9 Hz, 1H, H-2 phenothiazine), 7.20–7.10 (m, 4H, 2 -CH=CH-, 2H phenothiazine), 6.94–6.89 (m, 1H, H7 or H8 phenothiazine), 6.87–6.82 (m, 2H, phenothiazine), 3.86 (t, *J* = 7.2 Hz, 2H, NCH_2_), 3.60 (t, *J* = 6.4 Hz, 2H, OCH_2_), 2.76 (s, 3H, CH_3_), 1.87–1.78 (m, 2H), 1.56–1.42 (m, 4H), 1.41–1.34 (m, 2H), 0.89 (s, 9H, CH_3_), 0.04 (s, 6H, CH_3_). ^13^C NMR (100 MHz, CDCl_3_) *δ* 155.2 (C), 151.8 (C), 145.1 (C), 144.8 (C), 141.5 (C), 140.9 (C), 139.1 (C), 138.6 (C), 131.3 (C), 129.8 (CH), 129.0 (CH + C), 128.5 (CH), 128.4 (CH), 127.8 (CH), 127.5 (CH), 127.3 (CH), 126.4 (CH), 126.2 (CH), 125.9 (CH), 125.22 (CH), 125.20 (C), 124.2 (C), 122.6 (CH), 115.4 (CH), 115.3 (CH), 63.1 (CH), 47.5 (CH), 32.8 (CH), 26.9 (CH), 26.8 (CH), 26.0 (CH), 25.5 (CH), 24.3 (CH), 18.4 (C), −5.26 (CH). HRMS (ESI) calcd for C_41_H_48_N_3_OSSi [M+H]^+^ 658.3282, found 658.3287.

#### 3.8.4. 1-(6-((tert-Butyldimethylsilyl)oxy)hexyl)-1,2,3,4-tetrahydroquinoline-6-carbaldehyde (**1f’**)

The use compound **1f** (49 mg, 0.19 mmol), imidazole (36 mg, 0.53 mmol), *tert*-butyldimethylsilyl chloride (41 mg, 0.26 mmol) and DMF (0.2 mL) (reaction time 4 h) in general procedure afforded the title compound **1f′** (62 mg, 87%) as orange oil, *R_f_* 0.28 (10:1 hexane/EtOAc). IR (*ν*_max_, cm^−1^, KBr): 2931, 2894, 2568, 1674, 1599, 1558, 1521, 1472, 1439, 1412, 1348, 1322, 1255, 1185, 1151, 1104, 1006, 837, 809, 776. ^1^H NMR (400 MHz, CDCl_3_): *δ* 9.63 (s, 1H, CHO), 7.53 (dd, *J*= 8.7 Hz, 2.2 Hz, 1H, H-7 quinoline), 7.44 (d, *J* = 2.2 Hz, 1H, H-5 quinoline), 6.55 (d, *J* = 8.7 Hz, 1H, H-8 quinoline), 3.61 (t, *J* = 6.6 Hz, 2H, OCH_2_), 3.37 (t, *J* = 5.7 Hz, 2H, H-2 quinoline), 3.31 (t, *J* = 7.6 Hz, 2H, NCH_2_), 2.77 (t, *J* = 6.4 Hz, 2H, H-4 quinoline), 1.96–1.88 (m, 2H, 3-H quinoline), 1.65–1.61 (m, 2H, CH_2_), 1.55–1.51 (m, 2H, CH_2_), 1.42–1.35 (m, 4H, CH_2_), 0.89 (s, 9H, CH_3_), 0.05 (s, 6H, CH_3_). ^13^C NMR (100 MHz, CDCl_3_) *δ* 189.8 (CH), 150.2 (C), 131.1 (CH), 130.3 (CH), 124.5 (C), 121.5 (C), 109.2 (CH), 62.9 (CH), 51.3 (CH), 49.7 (CH), 32.7 (CH), 27.9 (CH), 26.8 (CH), 26.3 (CH), 25.9 (CH), 25.6 (CH), 21.4 (CH), 18.3 (C), −5.4 (CH).

### 3.9. General Procedure for Synthesis of **7a–e** and **3′**

A mixture of compound **6** or **3**, SeO_2_ and dioxane (2 mL) was stirred under an argon flow at 95–100 °C for 1–4 h and cooled to room temperature. The solvent was evaporated on a rotary evaporator. The residue was purified using column chromatography on silica gel (eluent CH_2_Cl_2_—EtOAc 100:1 or petroleum ether/EtOAc, gradient from 50:1 to 25:1) to give **7a–e** and **3′**.

#### 3.9.1. (E)-6-(4-(Bis(2-((tert-butyldimethylsilyl)oxy)ethyl)amino)styryl)-3-phenylquinoxaline-2-carbaldehyde (**7b**)

The use of compound **6b** (100 mg, 0.15 mmol), SeO_2_ (20 mg, 0.18 mmol) and dioxane (2 mL) (reaction time 1 h) in general procedure afforded the title compound **7b** (70 mg, 69%) as dark red powder. Mp 130–132 °C (hexane), *R_f_* 0.45 (4:1 hexane/EtOAc). IR (KBr, *ν*_max_/cm^−1^): 2954 (CH), 2928 (CH), 2883 (CH), 1715 (C=O), 1588 (C-N, C=C), 1518, 1481, 1471, 1449, 1398, 1352, 1294, 1254, 1188, 1133, 1106, 1071, 900, 830. ^1^H NMR (500 MHz, CDCl_3_) *δ* 10.29 (1H, s, CH=O), 8.21 (d, *J* = 8.9 Hz, 1H, H-8 quinoxaline), 8.10 (d, *J* = 1.8 Hz, 1H, H-5 quinoxaline), 8.07 (dd, *J* = 8.9, 1.8 Hz, 1H, H-7 quinoxaline), 7.72–7.69 (m, 2H, *o*-Ph), 7.58–7.54 (m, 3H, *m,p*-Ph), 7.45 (d, *J* = 8.9 Hz, 2H, H-3,5-aniline), 7.35 (d, *J* = 16.2 Hz, 1H, -HC=CH-), 7.10 (d, *J* = 16.2 Hz, 1H, -HC=CH-), 6.72 (d, *J* = 8.9 Hz, 2H, H-2,6-aniline), 3.79 (t, *J* = 6.4 Hz, 4H, OCH_2_), 3.57 (t, *J* = 6.4 Hz, 4H, NCH_2_), 0.90 (s, 18H, CH_3_), 0.04 (s, 12H, CH_3_). ^13^C NMR (150 MHz, CDCl_3_) *δ* 191.1 (CH), 155.2 (C), 148.7 (C), 143.8 (C), 143.7 (C), 143.6 (C), 140.7 (C), 136.9 (C), 133.8 (CH), 130.3 (CH), 129.9 (CH), 129.7 (CH), 129.2 (CH), 128.7 (CH), 128.6 (CH), 125.0 (CH), 124.1 (C), 122.1 (CH), 111.7 (CH), 60.4 (CH), 53.6 (CH), 25.9 (CH), 18.3 (C), −5.3 (CH). HRMS (ESI) calcd for C_39_H_54_N_3_O_3_Si_2_ [M+H]^+^ 668.3698, found 668.3700. 

#### 3.9.2. (E)-N-(6-((tert-butyldimethylsilyl)oxy)hexyl)-N-ethyl)amino)styryl)-3-phenylquinoxaline-2-carbaldehyde (**7c**)

The use of compound **6c** (43 mg, 0.074 mmol), SeO_2_ (10 mg, 0.09 mmol) and dioxane (0.5 mL) (reaction time 1 h) in general procedure afforded the title compound **7c** (40 mg, 91%) as dark red oil, *R_f_* 0.54 (10:3 hexane/EtOAc). IR (KBr, *ν*_max_/cm^−1^): 2925 (CH), 2854 (CH), 1722 (C=O), 1595 (C-N, C=C), 1518, 1485, 1462, 1401, 1369, 1350, 1264, 1188, 1133, 1096, 1010, 888, 833. ^1^H NMR (600 MHz, CDCl_3_) *δ* 10.28 (1H, s, CH=O), 8.21 (d, *J* = 8.9 Hz, 1H, H-8 quinoxaline), 8.10 (s, 1H, H-5 quinoxaline), 8.07 (dd, *J* = 8.9, 1.8 Hz, 1H, H-7 quinoxaline), 7.73–7.67 (m, 2H, *o*-Ph), 7.59–7.53(m, 3H, *m,p*-Ph), 7.46 (d, *J* = 8.5 Hz, 2H, H-3,5-aniline), 7.35 (d, *J* = 16.0 Hz, 1H, -HC=CH-), 7.08 (d, *J* = 16.0 Hz, 1H, -HC=CH-), 6.67 (d, *J* = 8.5 Hz, 2H, H-2,6-aniline), 3.62 (t, *J* = 6.3 Hz, 2H, OCH_2_), 3.42 (q, *J* = 6.9 Hz, 2H, NCH_2_), 3.31 (t, *J* = 7.5 Hz, 2H, NCH_2_), 1.67–1.60 (m, 2H), 1.57–1.51 (m, 2H), 1.44–1.34 (m, 4H), 1.18 (t, *J* = 6.9 Hz, 3H), 0.91 (s, 9H, CH_3_), 0.06 (s, 6H, CH_3_). ^13^C NMR (150 MHz, CDCl_3_) *δ* 191.2 (CH), 155.2 (C), 148.6 (C), 143.8 (C), 143.5 (C), 140.7 (C), 136.9 (C), 134.0 (CH), 130.3 (CH), 129.9 (C), 129.8 (CH), 129.7 (CH), 129.2 (CH), 128.8 (CH), 128.6 (CH), 124.8 (CH), 123.6 (C), 121.7 (CH), 111.7 (CH), 63.1 (CH), 50.4 (CH), 45.1 (CH), 32.9 (CH), 27.6 (CH), 27.0 (CH), 26.0 (CH), 25.8 (CH), 18.4 (C), 12.4 (CH), −5.2 (CH). HRMS (ESI) calcd for C_37_H_48_N_3_O_2_Si [M+H]^+^ 594.3510, found 594.3496. 

#### 3.9.3. (E)-6-(2-(10-(6-((tert-Butyldimethylsilyl)oxy)hexyl)-10H-phenothiazin-3-yl)vinyl)-3-phenylquinoxaline-2-carbaldehyde (**7e**)

The use of compound **6e** (17 mg, 0.026 mmol), SeO_2_ (3 mg, 0.03 mmol) and dioxane (0.5 mL) (reaction time 1 h) in general procedure afforded the title compound **7e** (14 mg, 82%) as dark red powder. Mp 132–133 °C, *R_f_* 0.37 (10:2 hexane/EtOAc). IR (KBr, *ν*_max_/cm^−1^): 2929 (CH), 2855 (CH), 1721 (C=O), 1596 (C=N, C=C), 1522, 1463, 1403, 1349, 1288, 1250, 1197, 1099, 1011, 888, 833. ^1^H NMR (500 MHz, CDCl_3_) *δ* 10.28 (s, 1H, CHO), 8.10 (d, *J* = 1.7 Hz, 1H, H-5 quinoxaline), 8.23 (d, *J* = 8.8 Hz, 1H, H-8 quinoxaline), 8.04 (dd, *J* = 8.8, 1.8 Hz, 1H, H-7 quinoxaline), 7.74–7.67 (m, 2H, *o*-Ph), 7.59–7.52 (m, 3H, *m,p*-Ph), 7.25 (s, 1H, H-4 phenothiazine), 7.37–7.28 (m, 3H), 7.18–7.11 (m, 3H), 6.96–6.90 (m, 1H, H7 or H8 phenothiazine), 6.88–6.82 (m, 2H, phenothiazine), 3.86 (t, *J* = 7.1 Hz, 2H, NCH_2_), 3.59 (t, *J* = 6.4 Hz, 2H, OCH_2_), 1.87–1.78 (m, 2H), 1.56–1.43 (m, 4H), 1.42–1.35 (m, 2H), 0.89 (s, 9H, CH_3_), 0.04 (s, 6H, CH_3_). ^13^C NMR (100 MHz, CDCl_3_) *δ* 191.1 (CH), 155.2 (C), 145.7 (C), 144.5 (C), 144.0 (C), 143.6 (C), 142.6 (C), 140.8 (C), 136.7 (C), 132.2 (CH), 130.8 (C), 130.4 (CH), 129.83 (CH), 129.81 (CH), 129.0 (CH), 128.6 (CH), 127.5 (CH), 127.4 (CH), 126.7 (CH), 126.0 (CH), 125.5 (CH), 125.3 (C), 125.2 (CH), 124.1 (C), 122.7 (CH), 115.5 (CH), 115.3(CH), 63.1 (CH), 47.6 (CH), 32.7 (CH), 26.9 (CH), 26.7 (CH), 26.0 (CH), 25.5 (CH), 18.4 (C), −5.26 (CH). HRMS (ESI) calcd for C_41_H_46_N_3_O_2_SSi [M+H]^+^ 672.3075, found 672.3072. 

#### 3.9.4. 6-Bromo-3-phenylquinoxline-2-carbaldehyde (**3′**)

The use of compound **3** (300 mg, 1 mmol), SeO_2_ (133 mg, 1.2 mmol) and dioxane (5 mL) (reaction time 4 h, temperature 95 °C) in general procedure afforded the title compound **3′** (281 mg, 90%) as yellow powder. Mp 147–148 °C, *R_f_* 0.30 (10:1 hexane/EtOAc). IR (KBr, *ν*_max_/cm^−1^): 1714 (C=O), 1595, 1526, 1449, 1389, 1348, 1201, 1052, 1009, 922, 877, 830, 819. ^1^H NMR (400 MHz, CDCl_3_): *δ* 10.29 (s, 1H, C(O)H), 8.39 (d, *J* = 2.0 Hz, 1H, H-8 quinoxaline) 8.16 (d, *J* = 8.9 Hz, 1H, H-5 quinoxaline), 7.93 (dd, *J* = 8.9 Hz, 2.0 Hz, 1H, H-6 quinoxaline), 7.72–7.66 (m, 2H, *o*-Ph), 7.60–7.51 (m, 3H, *m, p*-Ph). ^13^C NMR (100 MHz, CDCl_3_): *δ* 190.8 (CH), 155.3 (C), 145.2 (C), 143.2 (C), 139.7 (C), 136.1 (C), 134.6 (CH), 131.7 (CH), 131.4 (CH), 130.2 (CH), 129.9 (CH), 128.8 (CH), 128.7 (C). 

### 3.10. General Procedure for Chromophore Synthesis

A mixture of compounds **7** and **8** in dry ethanol was stirred at 60–70 °C for 1–6 h. The reaction mixture was cooled to room temperature, the solvent was evaporated on a rotary evaporator and the residue was purified using column chromatography on silica gel (eluent CH_2_Cl_2_—EtOAc 100:1 or petroleum ether—EtOAc, 10:1) to give the target chromophores. Then, dyes were heated in dry ethanol at 70 °C for 1–2 min, cooled and filtrated.

#### 3.10.1. 2-(4-((E)-2-(6-((E)-4-(bis(2-((tert-butyldimethylsilyl)oxy)ethyl)amino)styryl)-3-phenylquinoxline-2-yl)vinyl)-3-cyano-5-(4-cyclohexylphenyl)-5-methylfuran-2(5H)-ylidene)malononitrile (**Chr-An2**)

The use of compounds **7b** (34 mg, 0.05 mmol) and **8** (17 mg, 0.05 mmol) and ethanol (3 mL) (reaction time 3 h) in general procedure afforded the title compound **Chr-An2** (28 mg, 62%) as black powder. Mp 256 °C, *R_f_* 0.33 (1:1 hexane/EtOAc). IR (KBr, *ν*_max_/cm^−1^): 2951 (CH), 2926 (CH), 2853 (CH), 1583 (C-N, C=C), 1551, 1479, 1448, 1373, 1356, 1252, 1229, 1182, 1136, 1100, 1056, 908, 827. ^1^H NMR (600 MHz, CDCl_3_) *δ* 8.09–8.00 (m, 3H, H-7,8 quinoxaline, 1H -HC=CH-TCF), 7.97 (s, 1 H, H-5 quinoxaline), 7.59–7.53 (m, 1 H, *p*-Ph), 7.50–7.40 (m, 4H), 7.38–7.30 (m, 4H), 7.15 (d, *J* = 7.9 Hz, 2H, C_6_H_4_Cy), 7.06 (d, *J* = 16.1 Hz, 1H, -HC=CH- aniline), 7.00 (d, *J* = 7.9 Hz, 2H, C_6_H_4_Cy), 6.72 (d, *J* = 8.2 Hz, 2H, H-2,6-aniline), 3.79 (t, *J* = 6.5 Hz, 4H, OCH_2_), 3.57 (t, *J* = 6.5 Hz, 4H, NCH_2_), 2.55–2.48 (m, 1 H, Cy), 2.05 (s, 3H, CH_3_), 1.90–1.81 (m, 4H, Cy), 1.79–1.74 (m, 1 H, Cy), 1.44–1.33 (m, 4H, Cy), 1.31–1.23 (m, 1H, Cy), 0.90 (s, 18H, CH_3_), 0.05 (s, 12H, CH_3_). ^13^C NMR (150 MHz, CDCl_3_) *δ* 175.1 (C), 172.7 (C), 155.8 (C), 150.7 (C), 148.8 (C), 143.2 (C), 143.0 (CH), 142.9 (C), 141.5 (C), 137.2 (C), 133.7 (CH), 132.2 (C), 129.8 (CH), 129.4 (2CH), 129.3 (CH), 128.9 (CH), 128.7 (CH), 128.0 (CH), 125.7 (CH), 124.9 (CH), 124.1 (C), 122.1 (CH), 120.2 (CH), 111.8 (CH), 111.3 (C), 110.7 (C), 109.7 (C), 103.3 (C), 99.4 (C), 60.3 (CH), 59.0 (C), 53.5 (CH), 44.2 (CH), 34.2 (CH), 34.1 (CH), 26.7 (CH), 25.9 (CH), 24.1 (CH), 18.3 (C), −5.3 (CH). HRMS (ESI) calcd for C_61_H_73_N_6_O_3_Si_2_ [M+H]^+^ 993.5277, found 993.5270. 

#### 3.10.2. 2-(4-((E)-2-(6-((E)-4-((6-((tert-butyldimethylsilyl)oxy)hexyl)(ethyl)amino)styryl)-3phenylquinoxalin-2-yl)vinyl)-3-cyano-5-(4-cyclohexylphenyl)-5-methylfuran-2(5H)ylidene) malononitrile (**Chr-An3**)

The use of compounds **7c** (28 mg, 0.047 mmol) and **8** (16 mg, 0.047 mmol) and ethanol (0.5 mL) (reaction time 1 h) in general procedure afforded the title compound **Chr-An3** (23 mg, 59%) as black powder. Mp 183–185 °C, *R_f_* 0.38 (10:3 hexan/EtOAc). IR (KBr, *ν*_max_/cm^−1^): 2926 (CH), 2852 (CH), 2229 (C=CN), 1584 (C-N, C=C), 1511, 1479, 1429, 1372, 1304, 1248, 1185, 1139, 1098, 1018, 954, 827. ^1^H NMR (400 MHz, CDCl_3_) *δ* 8.05–8.02 (m, 2H, H-7,8 quinoxaline), 8.00 (d, *J* = 15.7 Hz, 1H, -HC=CH-TCF), 7.93 (s, 1 H, H-5 quinoxaline), 7.60–7.53 (m, 1 H, *p*-Ph), 7.50–7.43 (m, 4H), 7.37–7.31 (m, 4H), 7.14 (d, *J* = 8.4 Hz, 2H, C_6_H_4_Cy), 7.03 (d, *J* = 16.1 Hz, 1H, -HC=CH- aniline), 6.99 (d, *J* = 8.4 Hz, 2H, C_6_H_4_Cy), 6.68 (d, *J* = 8.9 Hz, 2H, H-2,6-aniline), 3.62 (t, *J* = 6.3 Hz, 2H, OCH_2_), 3.43 (q, *J* = 7.0 Hz, 2H, NCH_2_), 3.31 (t, *J* = 7.5 Hz, 2H, NCH_2_), 2.55–2.46 (m, 1 H, Cy), 2.04 (s, 3H, CH_3_), 1.90–1.81 (m, 4H, Cy), 1.80–1.74 (m, 1 H, Cy), 1.69–1.60 (m, 2H), 1.58–1.51 (m, 2H), 1.46–1.33 (m, 8H), 1.31–1.23 (m, 1H, Cy), 1.20 (t, *J* = 6.9 Hz, 3H), 0.91 (s, 9H, CH_3_), 0.06 (s, 6H, CH_3_). ^13^C NMR (100 MHz, CDCl_3_) *δ* 175.0 (C), 172.7 (C), 155.8 (C), 150.7 (C), 148.6 (C), 143.2 (C), 143.0 (C), 142.89 (CH), 142.87 (C), 141.5 (C), 137.2 (C), 133.8 (CH), 132.2 (C), 129.8 (CH), 129.4 (CH), 129.3 (CH), 128.8 (2CH), 128.0 (CH), 125.7 (CH), 124.7 (CH), 123.5 (C), 121.7 (CH), 120.1 (CH), 111.7 (CH), 111.3 (C), 110.7 (C), 109.7 (C), 103.2 (C), 99.3 (C), 63.1 (CH), 58.9 (C), 50.4 (CH), 45.0 (CH), 44.2 (CH), 34.2 (CH), 34.1 (CH), 32.8 (CH), 27.6 (CH), 27.0 (CH), 26.7 (CH), 26.0 (CH), 25.8 (CH), 24.1 (CH), 18.4 (C), 12.4 (CH), −5.3 (CH). HRMS (ESI) calcd for C_59_H_67_N_6_O_2_Si [M+H]^+^ 919.5089, found 919.5080. 

#### 3.10.3. 2-(4-((E)-2-(6-((E)-2-(10-(6-((tert-butyldimethylsilyl)oxy)hexyl)-10H-phenothiazin-3-yl)vinyl)-3-phenylquinoxalin-2-yl)vinyl)-3-cyano-5-(4-cyclohexylphenyl)-5-methylfuran-2(5H)-ylidene)malononitrile (**Chr-PT**)

The use of compounds **7e** (25 mg, 0.037 mmol) and **8** (13 mg, 0.037 mmol) and ethanol (0.3mL) (reaction time 4.5 h) in general procedure afforded the title compound **Chr-PT** (30 mg, 81%) as dark red powder. Mp 203–204 °C, *R_f_* 0.36 (10:3 hexan/EtOAc). IR (KBr, *ν*_max_/cm^−1^): 2929 (CH), 2855 (CH), 2229 (C=CN), 1596 (C-N, C=C), 1512, 1469, 1430, 1371, 1306, 1248, 1187, 1099, 1017, 950, 826. ^1^H NMR (400 MHz, CDCl_3_) *δ* 8.11 (d, *J* = 8.8 Hz, 1H, H-8 quinoxaline), 8.07 (d, *J* = 15.7 Hz, 1H, -CH=CH-TCF), 8.03 (d, *J* = 8.8 Hz, 1H, H-7 quinoxaline), 8.02 (s, 1H, H-5 quinoxaline), 7.60–7.55 (m, 1 H, *p*-Ph), 7.50–7.45 (m, 2H, *m*-Ph), 7.40–7.31 (m, 5H), 7.27 (d, *J* = 16.2 Hz, 1H, -CH=CH-phenothiazine), 7.18–7.11 (m, 5H), 6.99 (d, *J* = 8.4 Hz, 2H, C_6_H_4_Cy), 6.96–6.90 (m, 1H, H7 or H8 phenothiazine), 6.88–6.83 (m, 2H, phenothiazine), 3.87 (t, *J* = 7.2 Hz, 2H, NCH_2_), 3.56 (t, *J* = 6.4 Hz, 2H, OCH_2_), 2.55–2.46 (m, 1 H, Cy), 2.05 (s, 3H, CH_3_), 1.90–1.73 (m, 5H, Cy), 1.55–1.34 (m, 12H), 1.31–1.23 (m, 1H, Cy), 0.88 (s, 9H, CH_3_), 0.03 (s, 6H, CH_3_).^13^C NMR (100 MHz, CDCl_3_) *δ* 175.0 (C), 172.7 (C), 155.8 (C), 150.7 (C), 145.7 (C), 144.4 (C), 143.6 (C), 143.0 (C), 142.7 (C), 141.8 (C), 141.6 (C), 137.0 (CH), 132.1 (C), 131.9 (CH), 130.7 (C), 129.9 (CH), 129.7 (CH), 129.5 (CH), 129.4 (CH), 129.1 (CH), 128.9 (CH), 128.0 (C), 127.43 (CH), 127.38 (CH), 126.7 (CH), 126.0 (CH), 125.7 (CH), 125.3 (CH), 125.2 (CH), 124.0 (C), 122.7 (CH), 120.6 (CH), 115.4 (CH), 115.3 (CH), 111.2 (C), 110.6 (C), 109.6 (C), 103.6 (C), 99.4 (C), 63.0 (CH), 59.1 (C), 47.6 (CH), 44.2 (CH), 34.2 (CH), 34.1 (CH), 32.7 (CH), 26.8 (CH), 26.70 (CH), 26.68 (CH), 26.0 (CH), 25.5 (CH), 24.1 (CH), 18.3 (C), −5.3 (CH). HRMS (ESI) calcd for C_63_H_65_N_6_O_2_SSi [M+H]^+^ 997.4653, found 997.4667. 

#### 3.10.4. 2-(4-((E)-2-(6-((E)-2-(1-(6-((tert-Butyldimethylsilyl)oxy)hexyl)-1,2,3,4-tetrahydroquinolin-6-yl)vinyl)-3-phenylquinoxalin-2-yl)vinyl)-3-cyano-5-(4-cyclohexylphenyl)-5methylfuran-2(5H)-ylidene)malononitrile (**Chr-TQ**)

The use of compounds **7f** (10.0 mg, 0.016 mmol) and **8** (6.2 mg, 0.018 mmol) and ethanol (0.5 mL) (reaction time 6 h, temperature 60 °C) in general procedure afforded the title compound **Chr-TQ** (10 mg, 54%) as black powder, mp 146–147 °C, *R_f_* 0.33 (4:1 hexane/EtOAc. IR (KBr, ν, cm^–1^): 2929, 2851, 1584, 1510, 1478, 1373, 1346, 1308, 1188, 1103, 1017, 958, 899, 834, 803, 775. ^1^H NMR (400 MHz, CDCl_3_): *δ* 8.04 (d, *J* = 8.7 Hz, 1H, H-5 quinoxaline), 8.02 (d, *J* = 15.5 Hz, 1H, -CH=CH-), 8.17 (dd, *J* = 8.7 Hz, 1.6 Hz, 1H, H-6 quinoxaline), 7.93 (d, *J* = 1.6 Hz, 1H, H-8 quinoxaline), 7.59–7.51 (m, 1H, *p*-Ph), 7.50–7.43 (m, 2H, *m*-Ph), 7.37–7.33 (m, 2H, *o*-Ph), 7.32 (d, *J* = 15.5 Hz, 1H, -CH=CH-), 7.29 (d, *J* = 16.4 Hz, 1H, -CH=CH-), 7.27 (dd, *J* = 8.6, 2.0 Hz, 1H, H-7 quinoline), 7.20 (d, *J* = 2.0 Hz, 1H, H-5 quinoline), 7.14 (d, *J* = 8.4 Hz, 2H, C_6_H_4_Cy), 7.01 (d, *J* = 16.4 Hz, 1H, -CH=CH-), 6.99 (d, *J* = 8.4 Hz, 2H, C_6_H_4_Cy), 6.56 (d, *J* = 8.7 Hz, 1H, H-8 quinoline), 3.61 (t, *J* = 6.4 Hz, 2H, OCH_2_), 3.34 (t, *J* = 5.6 Hz, 2H, H-2 quinoline), 3.29 (t, *J* = 7.5 Hz, 2H, NCH_2_), 2.79 (t, *J* = 6.3 Hz, 2H, H-4 quinoline), 2.50 (m, 1H, PhCH<), 2.03 (s, 3H, CH_3_), 1.97 (p, *J* = 6.3, 5.6 Hz, 2H, H-3 quinoline), 1.90–1.99 (m, 4H, CH_2_-Cy), 1.76 (m, 1H, CH_equ_), 1.65–1.59 (m, 2H, CH_2_), 1.56–1.53 (m, 2H, CH_2_), 1.44–1.35 (m, 8H, CH_2_, CH_2_-Cy), 1.21 (m, 1H, CH_axi_), 0.90 (s, 9H, CH_3_), 0.06 (s, 6H, CH_3_). ^13^C NMR (100 MHz, CDCl_3_): δ 175.1 (C), 172.8 (C), 155.9 (C), 150.7 (C), 146.3 (C), 143.3 (C), 143.1 (C), 142.9 (CH), 142.8 (C), 141.5 (C), 137.3 (C), 134.0 (CH), 132.2 (C), 129.8 (CH), 129.4 (2CH), 129.3 (CH), 128.8 (CH), 128.0 (CH), 127.9 (CH), 127.2 (CH), 125.7 (CH), 124.6 (CH), 123.5 (C), 122.3 (C), 121.4 (CH), 120.0 (CH), 111.4 (C), 110.7 (C), 110.4 (CH), 109.7 (C), 103.2 (C), 99.3 (C), 63.1 (CH), 58.9 (C), 51.4 (CH), 49.6 (CH), 44.2 (CH), 34.1 (CH), 32.8 (CH), 28.3 (CH), 27.0 (CH), 26.7 (CH), 26.4 (CH), 26.0 (2CH), 25.8 (CH), 24.2 (CH), 22.0 (CH), 18.4 (C), −5.3 (CH). HRMS (ESI) calcd for C_60_H_66_N_6_O_2_Si 931.5089, found 931.5111.

### 3.11. Computational Details

We performed a conformational search for the proposed chromophores in gas phase in the energy window 5 kcal/mole with an OPLS4 force field [51] using the MacroModel program [52]. Conjugated π-electron bridges of the studied chromophores permit the existence of eight rotational isomers in accordance with the presence of three single rotable bonds (Appendix A). The notations of the isomers are in line with the rules formulated in [27]. The analysis of the results of the conformational search made it possible to identify the most stable and probable conformers; the reference chromophore **DBA-VQV-TCF_CyPh_** is shown in Appendix A. The notations of the dihedral angles between different chromophore fragments are given in Appendix A. The geometry of these conformers was fully optimized at the B3LYP/6-31G* level (Appendix A); this computational scheme is the most widely used for the estimation of the geometry of various compounds, in particular, organic NLO chromophores [53,54], and recent analysis has confirmed that it provides reliable results, especially for quite large molecular systems [55]. Electric properties (dipole moment and molecular polarizabilities) were calculated using the (TD)DFT at the M06-2X/aug-cc-pVDZ level, which was shown to provide reliable estimations of chromophore characteristics with heterocyclic fragments in π-conjugated bridges [54], in particular, those with quinoxaline moiety in π-electron bridges [56]. Calculations were performed using the Jaguar program package [52,57].

Atomistic modeling of composite polymethylmethacrylate (PMMA)-based materials with the studied chromophore guests with 25 and 40 wt.% content was performed with an OPLS4 force field (Appendix A) to reveal the isolating ability of bulky substituents in donor fragments with the Desmond program package [58] using a multistage simulation workflow (compressive protocol); molecular dynamics was performed at the last stage at 400 K during 50 ns. PMMA was modelled by 10 chains of 60 units. 

### 3.12. Film Fabrication, Poling and NLO Measurements

The composite polymer materials were fabricated with PMMA as the polymer matrix (T_g_ = 98 °C) and chromophores as guests from a 6–7% solution of the polymer in cyclohexanone (for materials containing **Chr-An1**, **Chr-A2** and **Chr-An3**) via spin-coating at 5000 rpm for 90 s. or from a 2% solution of the polymer in dibromomethane (for materials containing **Chr-PT**, **Chr-TQ** and **Chr-Car**) via spin-coating at 5000 rpm for 60 s. After casting, the samples were kept in a vacuum drying oven at room temperature for 10–16 h and then at 60 °C for 1.5 h to remove the residual solvent. Films were poled at the corona triode setup in the corona discharge field. The field was applied for 20 min (poling time) to the films heated to 100–110 °C, which is ~5 °C higher than T_g_. The quality of orientation was controlled using the absorption change in the UV–Vis spectra detected before and after poling (Appendix A), and characterized using the order parameter η (Appendix A), calculated by the following formula: *η = 1 − A/A*_0_*,* where A and A_0_ are the absorptions of the polymer films after and before poling. SHG was performed using the femtosecond amplified laser system, which allowed measuring the SHG intensity emitted by the sample without any micro-objective or another focusing system. The laser system produced pulses with the following parameters: the wavelength was 1028 nm, pulse repetition rate was 3 kHz, pulse duration was 200 fs, pulse energy was 164 µJ and mean power of the laser beam was 492 mW. The beam diameter of 3 mm resulted in the peak pulse intensity of about 11.6 GW/cm^2^. The SHG intensity was measured using an α-quartz crystal as a source of a reference signal (quartz nonlinear coefficient *d*_11,*q*_ = 0.45 pm/V). 

## 4. Conclusions

Novel D-π-A′-π-A chromophores with quinoxaline cores as auxiliary acceptors and various donor moieties (aniline, carbazole, phenothiazine and tetrahydroquinoline) containing bulky *tert*-butyldimethylsilyloxy groups and TCF acceptors with bulky cyclohexylphenyl substituents were synthesized, and their photo-physical and thermal properties were investigated. All chromophores are characterized by pronounced solvatochromism, with the sign of the solvatochromic shift depending on the relative polarity of the solvent; when passing from nonpolar solvents, such as dioxane, to those of moderate polarity, such as chloroform or dichloromethane, positive solvatochromism (up to 52 nm) is observed. Further increase in solvent polarity (acetonitrile) results in negative solvatochromic shift (up to 53 nm). In the series λ_max_(**Chr-TQ**) > λ_max_(**Chr-An3**) > λ_max_(**Chr-An1**) > λ_max_(**Chr-An2**) > λ_max_(**Chr-PT**) > λ_max_(**Chr-Car**), the hypsochromic shift in absorption maximum is observed when passing to each next member of the series. The studied chromophores exhibit high thermal stability; the decomposition temperatures, according to DSC T_d_, are in the range 171–270 °C. According to the theoretical estimations, **Chr-TQ** has the highest β_tot_ value (937 × 10^−30^ esu) among the studied chromophores. Atomistic modeling of composite materials with the studied chromophores as guests demonstrated that the presence of bulky substituents in the donor fragments prevents notable aggregation of chromophores, even at high chromophore content (40 wt.%). The nonlinear optical performance of guest–host materials with 25 and 40 wt.% of suggested chromophore content was studied using a second harmonic generation technique to give the NLO coefficient, *d*_33_ up to 52 pm/V. Thus, the introduction of TBDMSO groups permits increasing the chromophore content in the material with the growth in NLO response. The study of the long-term stability of the NLO response demonstrates that more than 90% of the *d*_33_ values are preserved during a year at room temperature.

## Data Availability

The data presented in this study are available in Appendix A.

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
