# Peer review of "Synthesis of D-π-A′-π-A Chromophores with Quinoxaline Core as Auxiliary Acceptor and Effect of Various Silicon-Substituted Donor Moieties on Thermal and Nonlinear Optical Properties at Molecular and Material Level"

_molecules, 2023, doi:10.3390/molecules28020531_

Round 1

Reviewer 1 Report

The article describes the synthesis of novel quinoxaline-based structures functionalized with silyl alkyl chains as donor groups. Synthesis and compound characterization were performed in detail. The compounds were then investigated in terms of nonlinear optical properties and analyzed by a variety of techniques. Presentation and schemes are fluent and clear.

The article is well suited for publication in the journal, pending an improvement in the discussion of the data. As the authors have already published about quinoxaline-based structures functionalized with acceptor and other groups, comments and comparison with properties of previous systems should be necessarily brought to the readers attention in the present article.

Acronyms should be fully expressed at first appearance in the article. For instance: TCF in abstract and main text, also TBDMS and TBDPS in the introduction. Also, TBDMSO and PMMA.

Minor language expressions should be corrected. For instance:

Abstract, first line: A Novel….(erase A); one line from last: ….chromophores content….  

First page, last line: derivatives…

Page 4:  2.5.1.((4-Vinylphenyl)azanediyl……..

Also correct compounds name in the following pages, for instance:  2.6.4.(. E)-6-(2-(1-(6-((tert…..       into     2.6.4.(E)-6-(2-(1-(6-((tert…….

In Scheme 1: reflux (not refluxe)

Page 15 line 1:….and their….?

In ref 12:  mercury(II)

Author Response

  1. Acronyms should be fully expressed at first appearance in the article. For instance: TCF in abstract and main text, also TBDMS and TBDPS in the introduction. Also, TBDMSO and PMMA.

We have added the explanation of the acronyms.

  1. Minor language expressions should be corrected. For instance:

Abstract, first line: A Novel….(erase A); one line from last: ….chromophores content….

First page, last line: derivatives…

Page 4: 2.5.1.((4-Vinylphenyl)azanediyl……..

Also correct compounds name in the following pages, for instance:  2.6.4.(. E)-6-(2-(1-(6-((tert…..into 2.6.4.(E)-6-(2-(1-(6-((tert…….

In Scheme 1: reflux (not refluxe)

Page 15 line 1:….and their….?

In ref 12: mercury(II)

We have corrected all mentioned misprints and invalid expressions.

Reviewer 2 Report

I look great potential of this paper for potential readers. Some minor changes must be however introduced before the publications. In particular:

The main subject ofthe article is connected with application of novel materials in the optoelectronics  and organic photovoltaics. So, last reviews in this area should be cited in the introduction [such as Scientiae Radices, 1, 3-25 (2022)].

The mobile phase for TLC experiments should be specified.

Do m.p.'s was determined after recrystallisation?

The quality of Figure 4 is vely low.

Dihedral angles listed in the Table 3 should be explained on the separate Figure.

Cartesian coordinates for all optimised structures should be collected in the Supplementary Material.

The application of the B3LYP/6-31G* level of theory should be justified using respective references in the recent literature.

Author Response

  1. The main subject of the article is connected with application of novel materials in the optoelectronics  and organic photovoltaics. So, last reviews in this area should be cited in the introduction [such as Scientiae Radices, 1, 3-25 (2022)].

We have added the reference to the recent review (2022) by Xu H., Elder D.L., Johnson L.E et al devoted to the progress in the field on synthesis and design of new organic NLO chromophores (ref. 31), we consider it more suitable than the one suggested by the Reviewer.

  1. The mobile phase for TLC experiments should be specified.

The remark of the Reviewer is not quite clear: after the description of general synthesis technique the data for each synthesized compound and it is pointed in the title of the subsection; thus all the data in each subsection are related just to the title compound, including the data obtained by TLC. For TLC the Rf , the eluent and the solvents ratio are given. We have unified this description.

  1. Do m.p.'s was determined after recrystallisation?

In most cases, melting temperature was determined after purification by column chromatography. In separate cases the additional recrystallization from hexane was performed (these data are added). The following phrase was introduced to subsection 2.10:

“Then dyes were heated in dry ethanol at 70 °Ð¡ for 1-2 min, cooled and filtrated.”

  1. The quality of Figure 4 is vely low.

All the figures are improved; besides, some corrections were made and the following passage is added:

“…there are two endo- and two exothermic peaks. As the first exothermic peak is low-intensive (close to the base line), it was not clear whether the corresponding temperature is the Td. To clarify this item, the chromophore Chr-TQ was heated up to 180 °C inside the DSC/TGA unit and the complete chromophore decomposition was confirmed by TLC. Thus the Chr-TQ manifests much lower thermal stability compared to that for all other five chromophores studied here, the difference in Td reaching almost 100 °C compared to the most stable chromophores (Chr-Car, Chr-PT, Chr-An2).”

  1. Dihedral angles listed in the Table 3 should be explained on the separate Figure.

The notations of dihedral angles are given in Figure S3.

  1. Cartesian coordinates for all optimised structures should be collected in the Supplementary Material.

We have presented the Cartesian coordinates for all studied chromophores in the Supplementary Materials.

  1. The application of the B3LYP/6-31G* level of theory should be justified using respective references in the recent literature.
    We have used conventional computational scheme – B3LYP/6-31G(d) – for the geometry optimization of the studied chromophores. The B3LYP density functional is the most widely used one, it provides a recognized choice for the estimation of the geometry of various organic compounds, the mean absolute error for bond lengths being less than 0.01 a.u. [ Rev. 2012, 112, 289]. As the structural parameters are much less sensitive to the choice of density functional and basis set than energies and properties, and keeping in mind that in many previous papers 6-31G(d) basis set has proven itself well for the structure calculations (see, for example, [J. Chem. Phys., 2008, 129, 044109; Acc. Chem. Res., 2014, 47, 3258–3265]), we assume that the use of this computational scheme is quite satisfactory. Recent analysis has confirmed this assumption that the use of basis set of DZ-quality provides reliable results [Angew. Chem. Int. Ed.2022,61,e202205735], especially considering that the sizes of the studied systems are quite large (more than 100 atoms).

 We have added the following phrase to the 2.11 Subsection (Computational details):

“This computational scheme is the most widely used one for the estimation of the geometry of various compounds, in particular, organic NLO chromophores [J. Chem. Phys., 2008, 129, 044109; Acc. Chem. Res., 2014, 47, 3258–3265]; recent analysis has confirmed that it provides reliable results, especially for quite large molecular systems [Angew. Chem. Int. Ed.2022,61,e202205735].”

The corresponding references are added to the List of references (47 and 49).

Reviewer 3 Report

 Alexey A. Kalinin and co-authors reported the a D-π-A-π-A chromophores with quinoxaline core as auxiliary acceptor and various donor moieties containing bulky tert-butyldimethylsilyloxy group and TCF acceptor with bulky cyclohexylphenyl substituent. In the current manuscript, the authors studied the effects of bulky TBDMS and TBDPS groups in the synthesized chromophores. The presence of bulky substituent in the donor fragment prevents notable aggregation of chromophores even at high chromophore content. The synthesized compounds have significant NLO activity that can be correlated with the previously published results from the same group. The compound without bulky tert-butyldimethylsilyloxy has less piezoelectric d33 (33 pm/V at 25%) than the compound  Chr-An3 which is having bulky silicon substituents (44 pm/V at 25%). So, this study is an extension of the study published by the same group earlier. Dyes and Pigments, 2021, 184, 108801; Russian Chemical Bulletin, 2022, 71, 1009) This manuscript can be accepted after addressing the following comments.

1.       In Section 2.2, the authors described the synthesis of compound 1e only, It is suggested to write the general method for the synthesis of compounds 1a, 1b, and 1c-f in two subsections.

2.       The compound’s name incircled in the image below should be Da-Df.

3.       Section 3.1, the synthetic method is poorly written and needs to be written carefully and systematically for easy readability.

Author Response

  1. 1. In Section 2.2, the authors described the synthesis of compound 1e only, It is suggested to write the general method for the synthesis of compounds 1a, 1b, and 1c-f in two subsections.

We decided not to describe the general procedure for the synthesis of compound 1 for two reasons:

  • only one of 5 compounds (1a-e) is new – 1e;
  • there are objective reasons that prevent us from offering a uniform description of the synthesis: various solvents were used, different order of reagents addition, different temperature and reaction time… At the end of Subsection 2.2 the necessary references to synthetic procedures are given.
  1. The compound’s name incircled in the image below should be Da-Df.

We have revised Scheme 1 and the caption, we hope that in the present form it became more clear.

  1. Section 3.1, the synthetic method is poorly written and needs to be written carefully and systematically for easy readability.

We have revised Section 3.1 and added the following passages:

“Synthetic approach given here providing aminostyrylquinoxalinylcarbaldehydes 7a-e with the Riley reaction at the last step made it possible to obtain a wide range of their derivatives both with a dialkylaniline donor moiety [29, 43] and with a carbazole or phenothiazine one. However, it turned out to be difficult to obtain aldehyde 7f using this approach due to the low conversion (~30%) at the final step, besides close Rf values of the product and the starting reagent made it difficult to isolate 7f by column chromatography.”

and

“All olefin derivatives 2, 4, 5, 6, 7 were isolated as E-isomers as shown in Scheme 1, and chromophores were isolated as E,E-isomers, as evidenced by 1H NMR (J-CH=CH- = ~ 16 Hz).

The signals from ortho- and meta-protons of phenyl group at quinoxaline moiety are shifted to higher field and para-proton of phenyl group resonates in the lower field (7.60-7.53 ppm) due to the shielding effect of the aryl substituent in TCF moiety. This indicates that in the chloroform solution, only one conformer for all compounds exists with close spatial arrangement of Ph and CyPh moieties, as shown in Figure S1.”
